# A Non-isotropic Time Series Diffusion Model with Moving Average Transitions

**Chenxi Wang** [1 2]   **Linxiao Yang** [2]   **Zhixian Wang** [1 2]   **Liang Sun** [2]   **Yi Wang** [1]

## Abstract

Diffusion models, known for their generative ability, have recently been adapted to time series analysis. Most pioneering works rely on the standard isotropic diffusion, treating each time step and the entire frequency spectrum identically. However, it may not be suitable for time series, which often have more informative low-frequency components. We empirically found that direct application of standard diffusion to time series may cause gradient contradiction during training, due to the rapid decrease of low-frequency information in the diffusion process. To this end, we proposed a novel time series diffusion model, MA-TSD, which utilizes the moving average, a natural low-frequency filter, as the forward transition. Its backward process is accelerable like DDIM and can be further considered a time series super-resolution. Our experiments on various datasets demonstrated MA-TSD's superior performance in time series forecasting and super-resolution tasks.

## 1. Introduction

Time series data is widely adopted in the real world. Extensive examples include electricity consumption in power systems, stock prices in financial markets, traffic flows in transportation systems, etc. Over the past decade, remarkable time series models have been developed with versatile deep neural networks to perform various time series analyses (Wang et al., 2024).

In recent years, the diffusion model (Ho et al., 2020) has risen as a shining generative model, showing remarkable performance for image and video synthesis. Such supercity in modeling complex data distributions also drives the community to seek how to adapt it to time series, and thus empower

time series analysis (Yang et al., 2024). So far, pioneer works have accommodated the diffusion model for time series forecasting (Rasul et al., 2021; Shen & Kwok, 2023; Li et al., 2022), missing value imputation (Tashiro et al., 2021; Alcaraz & Strodthoff, 2023), uncertainty quantification (Li et al., 2024) and so on.

Despite the initial success of these works, most of them still relied on the classical standard isotropic diffusion model, namely the Denosing Diffusion Probabilistic Model (DDPM) (Ho et al., 2020). It treats each time step independently and applies the same diffusion schedule. In the frequency domain, both low and high-frequency components are also degraded identically (see Figure 1). However, low-frequency components are usually more informative than high-frequency ones in time series analysis (Xu et al., 2024). We found that decreasing the low-frequency components identically to the high-frequency ones during the diffusion process may cause a drastic reduction of the essential time series information. It may further lead to contradictions on the gradient directions of DDPM at different diffusion steps, impeding the training convergence (see Section 3). Therefore, it's inappropriate to handle all the frequencies with the same diffusion process, and the classical design of the DDPM doesn't fully fit the inductive bias of time series data.

To tackle such inequality in the frequency domain of time series, we utilized moving average operation, a natural low-pass filter, to build a non-isotropic time series diffusion model, Moving Average Time Series Diffusion (MA-TSD). In the forward process, moving averages are set by small-to-large kernel sizes, gradually coarsening the time series until zero-frequency components. The corresponding transition matrices are no longer diagonal like standard diffusion models. A corresponding dataset-based noise schedule is provided alongside. Similar to Denosing Diffusion Implicit Models (DDIM) (Song et al., 2021a), we give an accelerable backward process with a customized strategy to select backward steps. Naturally, with the coarse-to-fine philosophy, the backward process of MA-TSD can also be viewed as time series super-resolution. Empirically, we show on diverse datasets that MA-TSD has outstanding performances over time series forecasting and super-resolution tasks.

**Contributions**: 1) We empirically disclosed the training

[1]Department of Electrical and Electronic Engineering, The University of Hong Kong, Hong Kong SAR [2]DAMO Academy, Alibaba Group, Hangzhou, China. Correspondence to: Yi Wang <yiwang@eee.hku.hk>.

*Proceedings of the 42$^{nd}$ International Conference on Machine Learning*, Vancouver, Canada. PMLR 267, 2025. Copyright 2025 by the author(s).

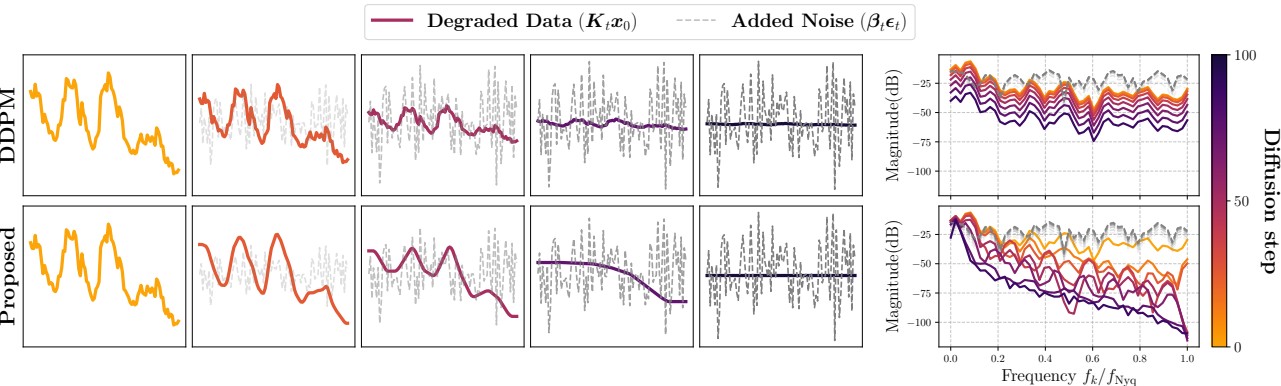

*Figure 1.* DDPM versus our proposed moving average diffusion process. Visualization relies on reparameterization, i.e. $\boldsymbol{x}_t = \boldsymbol{K}_t\boldsymbol{x}_0 + \beta_t\boldsymbol{\epsilon}_t$. Left: Comparison in time domain. Right: Comparison in the frequency domain. For illustration, $\boldsymbol{\epsilon}_t$ is fixed.

issue when directly applying DDPM to time series data, and explored the relationship between the gradient similarity at different diffusion steps and the change of frequency information. 2) We accordingly proposed a novel time series diffusion model with moving average as transition. The backward process can also be naturally considered as time series super-resolution. 3) We conducted extensive experiments to demonstrate our salient performances over existing DDPM-based diffusion models on time series-related tasks like time series forecasting and time series super-resolution.

## 2. Background

Given samples from a data distribution $q(\boldsymbol{x}_0)$, diffusion models are latent variable models in the form of $p_\theta(\boldsymbol{x}_0) = \int p_\theta(\boldsymbol{x}_{0:T}) \, d\boldsymbol{x}_{1:T}$, trying to approximate the unknown $q(\boldsymbol{x}_0)$. The joint distribution is usually modeled as a Markovian chain: $p(\boldsymbol{x}_{0:T}) = p(\boldsymbol{x}_T) \prod_{t=1}^{T} p_\theta(\boldsymbol{x}_{t-1}|\boldsymbol{x}_t)$. The latent variables of the diffusion models lie in the same space of the original data, i.e. $\boldsymbol{x}_t \in \mathbb{R}^L, \forall t \in [0, 1, \cdots, T]$. In our context, $\boldsymbol{x}_0$ is a time series with $L$ time steps.

The trainable parameters $\theta$ are optimized to minimize the negative variational lower bound of the log-likelihood on the data distribution $q(\boldsymbol{x}_0)$:

$$\min_\theta \mathcal{L} = \mathbb{E}_{q(\boldsymbol{x}_{0:T})}\left[\log q(\boldsymbol{x}_{1:T}|\boldsymbol{x}_0) - \log p_\theta(\boldsymbol{x}_{0:T})\right], \quad (1)$$

where the conditional joint $q(\boldsymbol{x}_{1:T}|\boldsymbol{x}_0)$ is the core of diffusion design. The designs of classical DDPM and DDIM will be introduced to pave the way for our proposed method.

### 2.1. Denosing diffusion probabilistic model

In DDPM, the forward process degrades an original data $\boldsymbol{x}_0$ by gradually compressing the data and adding Gaussian noises until $T \in \mathbb{N}^+$ diffusion steps so that all the structures of original data are lost, i.e. $q(\boldsymbol{x}_T|\boldsymbol{x}_0) = \mathcal{N}(\boldsymbol{0}, \boldsymbol{I})$. The

whole process is modeled as a Markovian chain:

$$q(\boldsymbol{x}_{1:T}|\boldsymbol{x}_0) := \prod_{t=1}^{T} q(\boldsymbol{x}_t|\boldsymbol{x}_{t-1}), \quad (2)$$

where the one-step transition is given by: $q(\boldsymbol{x}_t|\boldsymbol{x}_{t-1}) := \mathcal{N}\left(\sqrt{\alpha_t}\boldsymbol{x}_{t-1}, (1 - \alpha_t)\boldsymbol{I}\right)$. The coefficient $\alpha_t \in [0, 1]$ monotonically decreases with $t$. Through the property of Gaussian distribution, the transition from $\boldsymbol{x}_0$ to $\boldsymbol{x}_t$ can be derived:

$$q(\boldsymbol{x}_t|\boldsymbol{x}_0) = \mathcal{N}\left(\sqrt{\bar{\alpha}_t}\boldsymbol{x}_0, (1 - \bar{\alpha}_t)\boldsymbol{I}\right), \quad (3)$$

with the transition coefficient $\bar{\alpha}_t = \prod_{i=1}^{t} \alpha_i$. Through Bayes rule and the property of Markovian chain, Equation (2) can be reformulated as:

$$q(\boldsymbol{x}_{1:T}|\boldsymbol{x}_0) = q(\boldsymbol{x}_T|\boldsymbol{x}_0) \prod_{t=2}^{T} q(\boldsymbol{x}_{t-1}|\boldsymbol{x}_t, \boldsymbol{x}_0), \quad (4)$$

where the close form of $q(\boldsymbol{x}_{t-1}|\boldsymbol{x}_t, \boldsymbol{x}_0)$ can be analytically derived by Bayes rule. With the factorization of the forward $p_\theta(\boldsymbol{x}_{0:T})$ and the backward $q(\boldsymbol{x}_{1:T}|\boldsymbol{x}_0)$, the optimization target (Equation (1)) can be simplified. A large part of the simplified optimization target is $\min_\theta \mathbb{E}_{q(\boldsymbol{x}_0, \boldsymbol{x}_t)}[D_{\mathrm{KL}}(q(\boldsymbol{x}_{t-1}|\boldsymbol{x}_t, \boldsymbol{x}_0)\|p_\theta(\boldsymbol{x}_{t-1}|\boldsymbol{x}_t))]$. Therefore, the backward one-step transition $p_\theta(\boldsymbol{x}_{t-1}|\boldsymbol{x}_t)$ is modeled to approximate the true posterior:

$$p_\theta(\boldsymbol{x}_{t-1}|\boldsymbol{x}_t) = q(\boldsymbol{x}_{t-1}|\boldsymbol{x}_t, f_\theta(\boldsymbol{x}_t, t)), \quad (5)$$

where $f_\theta(\boldsymbol{x}_t, t)$ is a trainable denosing neural network to estimate the $\boldsymbol{x}_0$. The loss function can be consequently simplified as:

$$\mathcal{L} = \mathbb{E}_{\boldsymbol{x}_0 \sim q(\boldsymbol{x}_0), t \sim [1, T]}\left[\|f_\theta(\boldsymbol{x}_t, t) - \boldsymbol{x}_0\|_2^2\right], \quad (6)$$

with $\boldsymbol{x}_t = \sqrt{\bar{\alpha}_t}\boldsymbol{x}_0 + \sqrt{1 - \bar{\alpha}_t}\boldsymbol{\epsilon}, \boldsymbol{\epsilon} \sim \mathcal{N}(\boldsymbol{0}, \boldsymbol{I})$ via reparameterization. This means that the network $f_\theta$ takes in the noisy

data $\boldsymbol{x}_t$ and the current diffusion step $t$, and outputs the prediction of the clean data $\hat{\boldsymbol{x}}_0$. Alternatively, the network $f_\theta$ can also be optimized to estimate the added noise $\boldsymbol{\epsilon}_t$, and then apply $\hat{\boldsymbol{x}}_0 = (\boldsymbol{x}_t - \sqrt{1 - \bar{\alpha}_t} f_\theta(\boldsymbol{x}_t, t))/\sqrt{\bar{\alpha}_t}$ to obtain the prediction of clean data. After training, one can simply generate a synthetic data sample by iteratively denoising a $\boldsymbol{x}_T \sim p(\boldsymbol{x}_T)$ with Equation (5).

## 2.2. Denosing diffusion implicit model

Based on DDPM, DDIM generalized the foward process to be non-Markovian and derived a new backward process. First, DDIM bypasses the DDPM design of $q(\boldsymbol{x}_t|\boldsymbol{x}_{t-1})$, and directly considers the conditional joint $q(\boldsymbol{x}_{1:T}|\boldsymbol{x}_0)$ in the form of Equation (4), where $q(\boldsymbol{x}_{t-1}|\boldsymbol{x}_t, \boldsymbol{x}_0)$ is specially designed to ensure that for all $t$, the transition $q(\boldsymbol{x}_t|\boldsymbol{x}_0)$ always matches Equation (3). In other words, the marginal is satisfied, $\int q(\boldsymbol{x}_{t-1}|\boldsymbol{x}_t, \boldsymbol{x}_0)q(\boldsymbol{x}_t|\boldsymbol{x}_0)d\boldsymbol{x}_t = q(\boldsymbol{x}_{t-1}|\boldsymbol{x}_0)$.

Since the conditional joint $q(\boldsymbol{x}_{1:T}|\boldsymbol{x}_0)$ is defined as the same form as DDPM's, the optimization target of DDIM can be identically factorized, and the backward one-step transition is also chosen as Equation (5). In this way, DDIM shares the identical loss function of DDPM.

For backward process, DDIM offers an accelerable inference option. Specifically, given an ascending subset of $[1, \cdots, T]$, denoted as $\{t_i\}_1^\tau, \forall t_i \in [0, T]$ with $\tau \leq T$, DDIM allows the following sampling scheme:

$$\boldsymbol{x}_{t_{i-1}} = \sqrt{\bar{\alpha}_{t_{i-1}}} f_\theta(\boldsymbol{x}_{t_i}, t_i) + \sqrt{1 - \bar{\alpha}_{t_{i-1}} - \eta_{t_i}^2} \cdot \frac{\boldsymbol{x}_{t_i} - \sqrt{\bar{\alpha}_{t_i}} f_\theta(\boldsymbol{x}_{t_i}, t_i)}{\sqrt{1 - \bar{\alpha}_{t_i}}} + \eta_{t_i}\boldsymbol{\epsilon},$$

where $\{\eta_{t_i}\}$ are hyperparameters. DDIM can generate reasonable synthetic data within 50 steps for images. Besides, when $\eta_{t_i} = 0$, also known as deterministic sampling, such scheme is considered as the numerical solution of a probability flow ordinary differential equation (ODE)(Song et al., 2021b).

## 2.3. Conditional diffusion for time series forecasting

Time series forecasting can be viewed as a conditional generation task. Given a look-back window $\boldsymbol{c} \in \mathbb{R}^H$ with $H$ time steps, we are aimed to predict the next $L$ steps, i.e. the target window $\boldsymbol{x}_0$. In other words, we are interested in the conditional distribution $q(\boldsymbol{x}_0|\boldsymbol{c})$. To include the guidance of the look-back window into the diffusion model, we can add the condition at each transition(Shen & Kwok, 2023):

$$p(\boldsymbol{x}_{0:T}|\boldsymbol{c}) = p(\boldsymbol{x}_T) \prod_{t=1}^T p_\theta(\boldsymbol{x}_{t-1}|\boldsymbol{x}_t, \boldsymbol{c}). \qquad (7)$$

Accordingly, the condition $\boldsymbol{c}$ is handled as another feature input to the denosing network, $f_\theta(\boldsymbol{x}_t, t, \boldsymbol{c}) \approx \boldsymbol{x}_0$. Other types of time series analysis tasks can also be modeled in this way, for example super-resolution (conditional on low-resolution time series).

## 3. Empirical Findings on Applying DDPM to Time Series

In this section, we discovered a training issue when directly applying DDPM on time series (see Appendix A for detailed settings). As depicted in the left of Figure 2, the training process of DDPM on a typical time series dataset, `Electricity`, experienced large variations, even in the beginning stage. As we introduced above, the DDPM is optimized to denoise the corrupted time series at all diffusion steps (see Equation (6)), but we observed that the directions of model gradients at different diffusion steps could be contradicted during training. When we specifically probed the $100^{\text{th}}$ training step (see the left down of Figure 2), the gradients of about the first $25\%$ diffusion steps show greater similarity, while they could be opposite with the rest $75\%$ diffusion steps, and vice versa. Since each data sample is corrupted to varying degrees during DDPM training, the averaged gradient of each mini-batch can accordingly show considerable variation with polarized per-sample gradients. Consequently, it may lead to the unstable optimization of DDPM on time series.

To further analyze the reason for this phenomenon, we first explored the change of the frequency information during the whole diffusion process (see the right up of Figure 2). We computed the spectral energy ratio between low-frequency components and high-frequency ones of the time series at each diffusion step. During the whole diffusion process, low-frequency energy was dominant at the early stage, and then steeply decreased until a turning point after which the corrupted time series barely had salient low-frequency information, i.e., they became almost noises. If we compare this energy ratio change with the gradient similarity with different diffusion steps (see the right of Figure 2). We found that the shift of gradient directions is highly aligned with the turning point of energy ratios. It implies that when DDPM is directly applied on time series, low-frequency information decays so steeply that there is a large discrepancy in the model's perception of the input data, with a few steps in the early diffusion being informative, whereas, in the middle and late diffusion, they are nearly noises. Therefore, to prevent the drastic decline of the energy ratio, we expect a time series diffusion model to have a gradual diffusion process that can keep more low-frequency information.

In the following sections, we will introduce our method. Based on the moving average, a natural low-pass filter, it keeps more essential time series information during diffusion, alleviates the gradient contradiction during training, and thus obtains a less fluctuate training process (see the

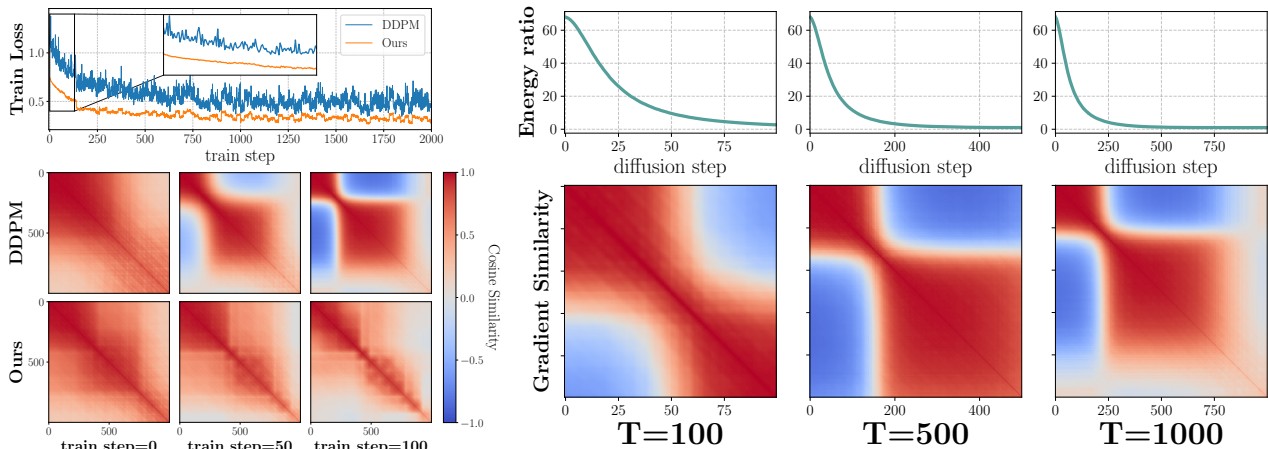

*Figure 2.* Gradient analysis on `Electricity` dataset. *Left up:* Training loss curves of DDPM and ours. *Left down:* Cosine similarity matrices of gradients w.r.t. the denosing network with different diffusion steps (total diffusion steps $T = 1000$). *Right up:* Energy ratio at different diffusion steps between the low-frequency components and the high-frequency ones. *Right down:* Cosine similarity matrices with different $T$ at $100^{\text{th}}$ training step.

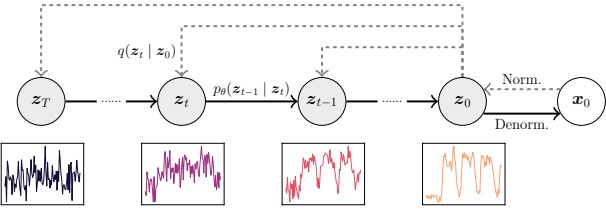

*Figure 3.* Our proposed MA-TSD. Dashed lines denote the forward process, while solid lines represent the backward process.

left of Figure 2).

# 4. Moving Average Time Series Diffusion

In this section, we present our proposed MA-TSD, depicted by Figure 3. Time series will first be normalized to be zero-mean and unit-variance. Then, we utilize moving average to build the diffusion model. Finally, the generated time series will be denormalized for downstream tasks.

## 4.1. Instance normalization

Since moving average operation doesn't modify the zero-frequency component of a time series, the final $p(\boldsymbol{x}_T)$ will no longer be a easy-to-sample standard Gaussian distribution if we directly construct moving average diffusion on $\boldsymbol{x}_0$. Therefore, we apply instance normalization to each target time series sample $\boldsymbol{x}_0 \in \mathbb{R}^L$, and obtain the normalized $\boldsymbol{z}_0 \in \mathbb{R}^L$ with zero mean and unit variance, specifically:

$$\boldsymbol{z}_0 = \frac{\boldsymbol{x}_0 - \mu(\boldsymbol{x}_0)}{\sigma(\boldsymbol{x}_0)}, \qquad (8)$$

where $\mu(\cdot), \sigma(\cdot)$ denotes the functions obtaining the mean and standard deviation of a time series, respectively. Therefore, we will build on the diffusion framework on the normalized time series. The denormalization strategies will be investigated in Section 4.4.

## 4.2. Non-isotropic forward process

We now consider the following transition process for normalized time series, $q(\boldsymbol{z}_t|\boldsymbol{z}_0) := \mathcal{N}\left(\boldsymbol{z}_t; \boldsymbol{K}_t\boldsymbol{z}_0, \beta_t^2\boldsymbol{I}\right)$, which can also be reparameterized to:

$$\boldsymbol{z}_t = \boldsymbol{K}_t\boldsymbol{z}_0 + \beta_t\boldsymbol{\epsilon}_t, \boldsymbol{\epsilon}_t \sim \mathcal{N}(\boldsymbol{0}, \boldsymbol{I}) \qquad (9)$$

with the *transition matrix* $\boldsymbol{K}_t \in \mathbb{R}^{L \times L}$, and the *noise schedule* $\beta_t \in \mathbb{R}$. In the standard DDPM, the transition matrix is diagonal with identical entries, $\boldsymbol{K}_t = \text{diag}(\sqrt{\bar{\alpha}_t})$, and the noise schedule is set accordingly to maintain the variance, $\beta_t = \sqrt{1 - \bar{\alpha}_t}$.

**Transition matrix.** In our design, we expect to utilize moving average to build our non-isotropic transition. First, let us consider non-overlapping moving average filters. The kernel sizes $\{k_i\}$ are naturally chosen as all the factors of the length of our target time series, given by:

$$\{k_i\}_1^n = \{k_i \in \mathbb{N} \mid 1 < k_i \le L, L \mod k_i = 0\}. \quad (10)$$

Here, we sort the $\{k_i\}_1^n$ in ascending order, and use the index $i$ instead of $t$ to tell from diffusion step indices for now.

The corresponding moving average kernels are denoted as $\{\hat{\boldsymbol{K}}_i\}_1^n$, and for each kernel, it convolves the normalized time series, i.e. $\hat{\boldsymbol{K}}_i * \boldsymbol{z}_0$. Such convolution can be unrolled and reformulated as matrix multiplication for generality,

namely $\hat{K}_i * z_0 = \bar{K}_i z_0$, with $\bar{K}_i = \text{Unroll}(\hat{K}_i) \in \mathbb{R}^{(L/k_i) \times L}$. We can further interpolate $\bar{K}_i$ along the time step axis to make it square to keep the shape of $\bar{K}_i z_0$ unchanged during transitions, and thus the transition matrix for $i^{\text{th}}$ moving average kernel is given as:

$$K'_i = \text{Interp}\left(\text{Unroll}(\hat{K}_i)\right) \in \mathbb{R}^{L \times L}. \quad (11)$$

Though the moving average transition matrices are defined well, we find that such transition could have large jumps between adjacent kernel sizes, since the factors of the time series length can be non-consecutive integers. To extend such design to the continuous case (i.e. arbitrary diffusion steps), we can further interpolate on $\{K'_i\}_1^n$ along side the diffusion steps to have unlimited $T$-step transition matrices $\{K_t\}_1^T$:

$$\{K_t\}_1^T = \text{Interp}\left(\{K'_i\}_1^n\right), \quad (12)$$

where $K_t$ is no longer a diagonal matrix like DDPM. A simple example of how we obtain the moving average transition is included in Appendix B.1.

**Noise schedule.** For the noise schedule, we follow the variance preserving principle in (Ho et al., 2020; Song et al., 2021b). The noise schedules of the standard diffusion can be analytically designed to keep variances according to the transition coefficient $\sqrt{\bar{\alpha}_t}$, i.e. $\beta_t = \sqrt{1 - \bar{\alpha}_t}$, while the decrease of time series variance caused by moving average varies by datasets. Therefore, we provide a dataset-based noise schedule to complement our forward process. Specifically, we can first compute the averaged decrease ratio $\gamma_t$ over the whole time series dataset:

$$\gamma_t = \mathbb{E}_{x_0 \sim q(x_0)}\left[\frac{\sigma(K_t x_0)}{\sigma(x_0)}\right]. \quad (13)$$

Then, we accordingly set $\beta_t = \sqrt{1 - \gamma_t^2}$ as our noise schedule to compensate for variance decrease. At the last diffusion step, the kernel size of the moving average equals to the time series length, and the corresponding $\gamma_T = 0, \beta_T = 1$, which ensures $q(z_T | z_0) = \mathcal{N}(0, I)$. It should be noted that though we conduct the diffusion model on the normalized time series, it makes no difference to compute $\{\gamma_t\}, \{\beta_t\}$ over the normalized or original time series, The proof can be found in the Appendix B.3.

**Conditional joint distribution.** Similar to DDIM, we directly define the following family of joint distribution: $q(z_{1:T} | z_0) := q(z_T | z_0) \prod_{t=2}^T q(z_{t-1} | z_t, z_0)$, where $q(z_T | z_0) = \mathcal{N}\left(z_T; K_T z_0, \beta_T^2 I\right)$, and for $t \geq 2$:

$$q(z_{t-1} | z_t, z_0) =$$
$$\mathcal{N}\left(K_{t-1} z_0 + \frac{\sqrt{\beta_{t-1}^2 - \eta_t^2}}{\beta_t}(z_t - K_t z_0), \eta_t^2 I\right). \quad (14)$$

The mean is chosen in order to guarantee that the modelling of the joint matches the marginal for all $t$, i.e. $\int q(z_{t-1} | z_t, z_0) q(z_t | z_0) dz_t = q(z_{t-1} | z_0)$. In other words, the choice of Equation (14) ensures $q(z_t | z_0) = \mathcal{N}\left(z_t; K_t z_0, \beta_t^2 I\right)$ for all $t$. The proof of such choice is included in Appendix B.3.

### 4.3. Accelerable backward process

Analogous to DDIM, we now define the backward process on the normalized time series as follows: $p_\theta(z_{t-1} | z_t) = q(z_{t-1} | z_t, f_\theta(z_t, t, c))$, where the denosing network $f_\theta(z_t, t, c)$ tries to predict the clean normalized time series. For generality, we include the possible condition $c$ as input of the denoising network for the conditional generation task, like time series forecasting. For unconditional tasks, we can simply set $c = \varnothing$.

For the normalized time series, we modeled $q(z_{1:T} | z_0)$ and $p_\theta(z_{t-1} | z_t)$ the same as DDIM, so it's natural to derive the similar optimization target as Equation (6) but in the normalized space:

$$\mathcal{L}_z = \mathbb{E}_{z_0, c \sim q(z_0, c), t \sim [1, T]}\left[\|f_\theta(z_t, t, c) - z_0\|_2^2\right], \quad (15)$$

After training, we can also accelerate the backward process as we introduced in Section 2.2, given as:

$$z_{t_{i-1}} = K_{t_{i-1}} f_\theta(z_{t_i}, t_i, c) +$$
$$\frac{\sqrt{\beta_{t_{i-1}}^2 - \eta_{t_i}^2}}{\beta_{t_i}}(z_{t_i} - K_{t_i} f_\theta(z_{t_i}, t_i, c)) + \eta_{t_i} \epsilon, \quad (16)$$

where $\{t_i\}_1^\tau$ is an ascending sub-sequence of $[1, \cdots, T]$. The detailed proof that the accelerated backward process doesn't essentially change the training objective can be found in the (Song et al., 2021a).

**Acceleration strategy.** Though the backward process can be fastened by selecting a subset of total diffusion steps, how this subset is chosen may cause performance differences. Here, we offer a reasonable sampling strategy based on our moving average forward process. Specifically, we recall that the diffusion transition matrices $\{K_t\}$ are obtained by interpolating the original moving average transition matrices $\{K'_i\}$. We consider shortening the backward process by finding those diffusion steps whose transition matrices are the closest to the original $\{K'_i\}$. For the $i^{\text{th}}$ original matrix $K'_i$, we search by:

$$t_i^* = \arg\min_t \|K_t - K'_i\|_2^2, \quad \text{s.t.} \quad K_t \in \{K_t\}_1^T. \quad (17)$$

Therefore, we can collect $n$ diffusion steps $\{t_i^*\}_1^n$, corresponding to the original non-overlapping moving average kernels, as our accelerated backward steps. We call such a strategy as *factor-only backward* since the selected backward steps are only related to the factors of the length of

the time series. When $\eta_t = 0, \forall t \in [1, \cdots, T]$, the backward process can also be viewed as a numerical solution to an ODE with Euler discretization. Further, if the function $\texttt{Interp}(\cdot)$ in Equation (12) interpolates evenly between $\boldsymbol{K}'_i, \boldsymbol{K}'_{i-1}, \forall i \geq 2$, the factor-only backward essentially solves that ODE with larger steps. Refer to Appendix B.3 for details.

**Backward as super-resolution.** As the forward process is designed as gradually coarsening the time series, the backward process can be spontaneously utilized for super-resolution. Here, we don't have to include the low-resolution time series as the condition input into MA-TSD, since the framework itself demonstrates the multi-resolution property.

To be specific, let us consider a coarse time series whose downscale rate is one of the factors of the original time series length. Utilizing the moving average transition matrix, we denote such coarse time series as $\boldsymbol{x}_i = \boldsymbol{K}'_i \boldsymbol{x}_0$ and denote the normalized one as $\boldsymbol{z}_i$. The super-resolution scale we expected is then exactly $k_i$. With Equation (17), we can locate such scale in the diffusion process, and accordingly choose a subset of total diffusion steps as $[1, \cdots, t_i^*]$. Then, we can start with $\boldsymbol{z}_{t_i^*} = \boldsymbol{z}_i$, and iteratively apply Equation (16) to obtain a super-resolution result of $\boldsymbol{z}_i$. Therefore, the backward process of MA-TSD can be viewed as super-resolution.

Compared with the diffusion-based models which take low-resolution inputs as conditions, the time complexity is decreased from $\mathcal{O}(nM)$ to $\mathcal{O}(M)$, where $n$ is the number of scales and $\mathcal{O}(M)$ is the complexity of training.

### 4.4. Denormalization

Given the denoised $\hat{\boldsymbol{z}}_0$ from Equation (16), we need to denormalize it to generate the final time series $\boldsymbol{x}_0$, i.e. $\hat{\boldsymbol{x}}_0 = \hat{\boldsymbol{z}}_0 \cdot \hat{\sigma} + \hat{\mu}$. In this section, the choice of $\hat{\mu}, \hat{\sigma}$ is considered different, depending on the downstream application of MA-TSD.

**Time series synthesis.** For the unconditional generation, we usually expect unlimited time series synthesis. Therefore, we can simply sample from the empirical distribution of time series means and standard deviations, $\hat{\mu} \sim q_{\text{emp}}(\mu(\boldsymbol{x_0})), \hat{\sigma} \sim q_{\text{emp}}(\sigma(\boldsymbol{x_0}))$. Given a time series dataset, we can easily obtain the empirical distributions and denormalize them.

**Time series forecasting.** For time series forecasting, the mean and standard deviation of the target window is vital for the prediction accuracy (Kim et al., 2021; Qin et al., 2024). It's improper to randomly sample from the empirical distribution to conduct denormalization. We consider to utilize the look-back window $\boldsymbol{c}$ to produce the $\hat{\mu}, \hat{\sigma}$ for the target window. To prevent training a separate statistics

prediction network for $\hat{\mu}, \hat{\sigma}$, we optimize both the denosing model $f_\theta$ and the statistics prediction model with parameters $\omega, g_\omega = \{g_\omega^\mu, g_\omega^\sigma\}$, in a hybrid manner. In the appendix, Figure 7 shows how these two networks work together for time series forecasting. Specifically, we modify the loss function on the normalized data (Equation (15)) into a hybrid case:

$$\mathcal{L}_{\text{hybrid}} = \lambda_{\boldsymbol{z}} \mathcal{L}_{\boldsymbol{z}} + \lambda_\mu \mathcal{L}_\mu + \lambda_\sigma \mathcal{L}_\sigma, \tag{18}$$

where we denote $\mathcal{L}_\mu = \mathbb{E}_{\boldsymbol{x}_0 \sim q(\boldsymbol{x_0})} \left[ \|g_\omega^\mu(\boldsymbol{c}) - \mu(\boldsymbol{x}_0)\|_2^2 \right]$, and $\mathcal{L}_\sigma = \mathbb{E}_{\boldsymbol{x}_0 \sim q(\boldsymbol{x_0})} \left[ \|g_\omega^\sigma(\boldsymbol{c}) - \sigma(\boldsymbol{x}_0)\|_2^2 \right]$. The coefficients $\lambda_{\boldsymbol{z}}, \lambda_\mu, \lambda_\sigma$ are hyperparameters. We interpret that $\mathcal{L}_{\boldsymbol{z}}$ aims to learn the *shape* of the target time series, while $\mathcal{L}_\mu$ and $\mathcal{L}_\sigma$ are set for the *statistics*. Therefore, we can set $\hat{\mu} = g_\omega^\mu(\boldsymbol{c}), \hat{\sigma} = g_\omega^\sigma(\boldsymbol{c})$, to denormalize $\boldsymbol{z}_0$ in the context of time series forecasting. Besides, when the coefficients $\lambda_{\boldsymbol{z}}, \lambda_\mu, \lambda_\sigma$ are set properly, we can prove that the hybrid loss (Equation (18)) is essentially the upper bound of the loss of conditional diffusion models without instance normalization. The detailed proof can be found in Appendix B.3.

**Time series super-resolution.** For super-resolution, since the coarse time series $\boldsymbol{x}_i$ obtained by moving average shares the same mean of the target time series, we can use $\hat{\mu} = \mu(\boldsymbol{x}_i)$. On the other hand, though the standard deviations are not shared, we can utilize the dataset-based noise schedule to re-scale $\sigma(\boldsymbol{x}_i)$ as: $\hat{\sigma} = \sigma(\boldsymbol{x}_i)/\gamma_{t_i^*}$ where $\gamma_{t_i^*}$ is the decrease ratio of the standard deviation at the diffusion step $t_i^*$ mentioned before.

## 5. Experiments

In this section, we mainly focus on two important time series analysis tasks, forecasting and super-resolution. The standard time series synthesis task is included in Appendix C.1. An ablation study of our framework design is also included.

### 5.1. Time series forecasting

**Datasets.** We consider six real-world datasets with diverse temporal dynamics, commonly used by the community (Wang et al., 2024), namely `Electricity`, `ETTh2`, `ETTm2`, `exchange`, `traffic`, `weather`.

**Evaluation metrics.** We assess time series forecasting using MSE (Mean Squared Error) for deterministic accuracy and CRPS (Continuous Ranked Probability Score) for probabilistic accuracy.

**Benchmarks.** We compare our proposed MA-TSD with other diffusion-based time series forecasting models, including CSDI (Tashiro et al., 2021), SSSD (Alcaraz & Strodthoff, 2023), D3VAE (Li et al., 2022), TMDM (Li et al., 2024) and mr-diff (Shen et al., 2024). Details about the implementation and comparison to non-diffusion models are included in the Appendix C.2.

*Table 1.* Average MSEs over prediction lengths $L = \{96, 192, 336, 720\}$. The best is **bold** and the second best is underlined.

| METHOD | ELECTRICITY | ETTh2 | ETTm2 | EXCHANGE | TRAFFIC | WEATHER | RANK |
|---|---|---|---|---|---|---|---|
| CSDI | 0.4581 | 0.2571 | 2.1230 | 1.2557 | 0.4991 | **0.1938** | 3.50 |
| SSSD | 1.0257 | 0.7201 | 0.8936 | 2.9004 | 1.9662 | 0.6905 | 5.17 |
| D3VAE | 0.8450 | 1.3961 | 3.3449 | 2.1086 | 6.3583 | 1.5461 | 5.67 |
| TMDM | 0.4071 | 0.2508 | 0.1789 | 0.7885 | 0.1805 | 0.2209 | 2.83 |
| MR-DIFF | 0.5287 | 0.2172 | 0.1700 | 0.4801 | 0.2471 | 0.2078 | 2.67 |
| **MA-TSD** | **0.3404** | **0.2121** | **0.1241** | **0.3718** | **0.1660** | 0.2074 | **1.17** |

*Table 2.* Average CRPSs over prediction lengths $L = \{96, 192, 336, 720\}$. The best is **bold** and the second best is underlined.

| METHOD | ELECTRICITY | ETTh2 | ETTm2 | EXCHANGE | TRAFFIC | WEATHER | RANK |
|---|---|---|---|---|---|---|---|
| CSDI | 0.1939 | 0.1638 | 0.4720 | 0.3028 | 0.1883 | **0.1261** | 3.33 |
| SSSD | 0.3216 | 0.3108 | 0.3565 | 0.6743 | 0.4579 | 0.3129 | 5.17 |
| D3VAE | 0.3111 | 0.4173 | 0.6497 | 0.5380 | 0.8314 | 0.4497 | 5.67 |
| TMDM | 0.1881 | 0.1591 | 0.1253 | 0.2959 | **0.1076** | 0.1380 | 2.00 |
| MR-DIFF | 0.2357 | 0.1647 | 0.1293 | **0.2117** | 0.1453 | 0.1506 | 3.17 |
| **MA-TSD** | **0.1747** | **0.1567** | **0.1135** | 0.2178 | 0.1156 | 0.1501 | **1.67** |

**Results.** As depicted in Table 1 and Table 2, the proposed MA-TSD generally outperforms the benchmark time series diffusion models, achieving the best or the second best position on 6/6 and 5/6 datasets, respectively. The improvement on the weather dataset is marginal, possibly because it is recorded in a higher resolution (Table 6). Both informative high-frequency components and stochastic noises are revealed, which could be simultaneously suppressed by the moving average process, and thus lead to the difficulty to accurately forecast. Refer to Appendix C.2 for visualization and Table 7 for the full results on each prediction length.

### 5.2. Time series super-resolution

**Datasets.** We consider three high-resolution real-world datasets with 5-minute resolution, MFRED, Wind, Solar. For each dataset, we test the models for the following 3 tasks, i.e. 5min-to-15min ($3\times$), 5min-to-30min ($6\times$), and 5min-to-60min ($12\times$).

**Evaluation metrics.** We assess time series super-resolution by Consistency and Context-FID. The former one measures MSE between the low-resolution inputs and the down-scaled super-resolution outputs (Saharia et al., 2022), while the latter one examines the quality of the super-resolution results compared to the real high-resolution time series (Jeha et al., 2022).

**Benchmarks.** We compare ours with two diffusion models directly conditioned on low-resolution inputs. One is trained under DDPM framework (Saharia et al., 2022), and the other is trained by flow matching with a variance-preserving path (Lipman et al., 2023), which we denote as FM-VP. The benchmark models are re-trained for each super-resolution scale, since the condition inputs change. Details about the implementation are all included in the Appendix C.3.

**Results.** Table 3 records the results of time series super-resolution. The proposed MA-TSD generally exceeds the conditional DDPM and FM-VP in terms of both consistency and quality. Despite the slight inferiority in Context-FID on the Solar dataset at the large scale, ours still outperformed the FM-VP model significantly in terms of consistency. Notably, MA-TSD performs SR naturally through its backward process instead of retraining individually on each scale. Besides, our SR backward starts in the middle of the whole process, resulting in even fewer backward steps compared to benchmarks (Figure 12). Therefore, we believe that our method provides a better trade-off of computational overheads, SR quality, and consistency to the low-resolution input. Refer to Appendix C.3 for more visualization.

### 5.3. Ablation study

In this section, we evaluate the effectiveness of important components and designs of MA-TSD through unlimited time series synthesis on the mentioned MFRED, Wind, Solar datasets.

**Key modules.** Two key different designs from the standard time series diffusion model are the moving average diffusion schedule and the instance normalization in our proposed MA-TSD. We compared the possible design with/without these two components, as shown in Table 4. Without MA and IN, namely directly applying DDIM on time series data, exhibited the worst performance. Equipped with either IN or

Table 3. Comparison on time series super-resolution. The best is **bold** and the second best is underlined.

| SCALE | METHOD | MFRED | | WIND | | SOLAR | |
|---|---|---|---|---|---|---|---|
| | | CONSIST. | CONTEXT-FID | CONSIST. | CONTEXT-FID | CONSIST. | CONTEXT-FID |
| 3 | **MA-TSD** | **0.0032** | **0.1047** | **0.0067** | **0.2863** | **0.0106** | **0.3491** |
| | DDPM | 0.0291 | 3.1028 | 0.0382 | 7.4843 | 0.0367 | 1.7197 |
| | FM-VP | 0.0328 | 1.3481 | 0.0636 | 4.2311 | 0.0523 | 0.7950 |
| 6 | **MA-TSD** | **0.0037** | **0.1235** | **0.0098** | **1.0241** | **0.0115** | 0.6972 |
| | DDPM | 0.0214 | 3.0740 | 0.0343 | 7.9524 | 0.0260 | 2.2293 |
| | FM-VP | 0.0238 | 1.3381 | 0.0505 | 4.4683 | 0.0361 | **0.6846** |
| 12 | **MA-TSD** | **0.0047** | **0.4358** | **0.0136** | **3.0567** | **0.0129** | 1.4135 |
| | DDPM | 0.0157 | 3.4044 | 0.0318 | 9.2504 | 0.0429 | 6.2596 |
| | FM-VP | 0.0156 | 1.5222 | 0.0350 | 4.8737 | 0.0249 | **0.8616** |

Table 4. Context-FIDs of MA-TSDs with different components. MA: Moving Average schedule. IN: Instance Normalization

| MODULES | | DATASETS | | |
|---|---|---|---|---|
| MA | IN | MFRED | WIND | SOLAR |
| - | - | 29.7093 | 52.3499 | 43.6236 |
| - | ✓ | 13.1704 | 44.6908 | 35.3704 |
| ✓ | - | 4.9403 | 21.3606 | 8.0744 |
| ✓ | ✓ | **2.6742** | **16.9026** | **8.0297** |

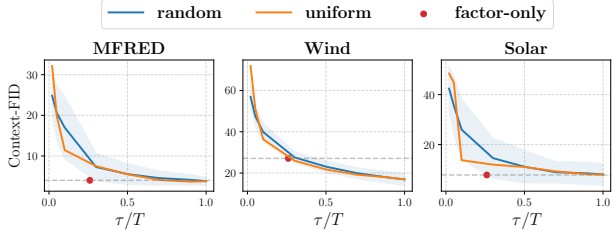

Figure 4. Comparison among accelerated backward strategies.

MA, the model obtained considerable improvements, with MA functioning more significantly than IN, which implies the importance of MA in our framework design. Combining both modules achieved the best scores over all three datasets.

**Accelerated backward.** For a given backward step budget $\tau \leq T = 100$, we compare our factor-only backward strategy with 1) randomly selecting subsets from $[1, \cdots, T]$ and 2) uniformly choosing a subset, i.e. $\texttt{linspace}(1, T, \tau)$. Notably, given a budget $\tau$, uniformly sampling renders a fixed subset $\{t_i\}_1^\tau$, resulting in a deterministic result.

As shown in Figure 4, given a fixed ratio $\tau/T$, randomly selecting the backward steps may cause large variances in model performances. Uniform sampling did offer a better result than random sampling, but the marginal gain on the performance decreases by $\tau/T$. Compared to both random and uniform strategies, our factor-only strategy consistently offers fair and effective results given the same $\tau/T$ budget, optimizing the resources while maintaining the quality of the results. Especially on MFRED and Solar, whose periodical patterns are more significant, the factor-only accelerating strategy could even achieve competitive performance with the full-step backward, indicating that the strategy captures the key transition steps and can be utilized for fastening MA-TSD.

## 6. Related Works

Diffusion models have been embraced by the time series analysis community for their advanced probability modeling ability. (Rasul et al., 2021) first combined autoregressive modeling of recurrent neural networks and the diffusion process for time series forecasting. Then (Shen & Kwok, 2023) proposed a non-autoregressive diffusion strategy for foreasting, improving on both efficiency and accuracy. In addition, there are also several works (Kollovieh et al., 2024; Alcaraz & Strodthoff, 2023; Tashiro et al., 2021) that link time series forecasting and time series imputation, modeling them with a conditional generation design, proposing unified frameworks for these two tasks with diffusion models.

Recently, the community began to fuse the unique time series property into diffusion models. (Fan et al., 2024) leveraged coarse time series data as guidance during the diffusion process, and added regularization terms into the loss function to constrain the backward process is coarse-to-fine. (Shen et al., 2024) set several diffusion stages, where the previous diffusion stage generates coarse time series as condition input for the latter stage to refine. (Liu et al., 2024) leveraged the historical windows to retrieve the $k$ nearest samples as references to guide diffusion model to generate more accurate forecasts. Despite the recent special design on time series data, they still rely on the DDPM process and

hardly improve on the typical isotropic design to meet the characteristics of time series.

Beyond time series diffusion models, some works similarly investigated non-isotropic diffusion models for images. (Hoogeboom & Salimans, 2023; Rissanen et al., 2023) designed frequency-domain diffusion with Gaussian blurring as transition. (Daras et al., 2023) tried to generalize the transition to linear corruptions, and gave examples of blurring and masking while (Bansal et al., 2024) proposed a diffusion model with arbitrary degradation functions, for example snowification and animorphosis, but without noise. However, few of them obtained tremendous improvements, let alone being further explored by our time series community.

Regarding diffusion design, probably the most related work to ours is (Hoogeboom & Salimans, 2023), which shared a similar high-level idea of building the degradation process with low-pass filters (blurring in theirs, MA in ours). However, they still tried to fit low-pass filters into the traditional DDPM's Markovian process in the frequency domain, while we reformulated a non-Markovian design with a new backward process. Besides, our noise schedule is specially designed to be dataset-based, regarding the variance decrease caused by the filters on different time series data. Please refer to Appendix B.4 for more detailed information.

## 7. Conclusion

In this paper, we first revealed that direct application of standard DDPM to time series data may cause gradient contradiction, because of rapid degradation of low-frequency information. A novel time series diffusion model, MA-TSD, is accordingly proposed, equipped with moving average forward transitions to keep more low-frequency information. The backward process can be accelerated in a DDIM style and further act as super-resolution. The experiments show that MA-TSD has superior performances over the state-of-the-art time series diffusion models in terms of forecasting and super-resolution.

## Software and Data

Our codes can be found in https://github.com/WillWang1113/Moving-Average-Diffusion.

## Acknowledgements

The work was supported in part by the Research Grants Council of the Hong Kong SAR (HKU 17200224), and in part by the Alibaba Group through Alibaba Research Intern Program.

## Impact Statement

This paper presents work whose goal is to advance the field of Machine Learning. There are many potential societal consequences of our work, none which we feel must be specifically highlighted here.

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

## A. Details of directly applying DDPM on time series

As an example of illustrating the potential glitch when DDPM is directly applied on time series data, we used `Electricity` dataset, a typical time series dataset with complex patterns. We consider $L = 400$ and the spectral energy ratio $e$ is calculated as follows:

$$e = \frac{\sum_{i=1}^{\zeta} |u_0^i|^2}{\sum_{i=\zeta+1}^{N_{\mathrm{Nyq}}} |u_0^i|^2}, \tag{19}$$

where we denote the real fast Fourier transform (rFFT) of the time series $\boldsymbol{x}_0$ as $\boldsymbol{u}_0 = [\boldsymbol{u}_0^1, \boldsymbol{u}_0^2, \cdots, \boldsymbol{u}_0^{N_{\mathrm{Nyq}}}]$, and $\boldsymbol{u}_0^i \in \mathcal{C}$ is the $i^{\mathrm{th}}$ frequency component. The parameter $\zeta$ is the split ratio. We set $\zeta = \mathrm{round}(0.2 \cdot N_{\mathrm{Nyq}})$ in this example, since the frequency components are in a relative low level after then (see Figure 5).

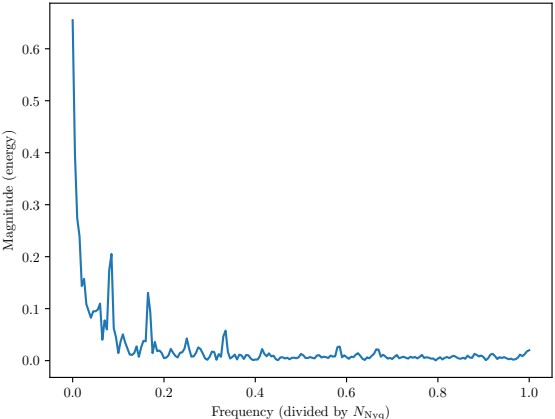

*Figure 5.* Frequency energy of a data sample in `Electricity` dataset. The x-axis is devided by $N_{\mathrm{Nyq}}$

During training, we fixed the random seed to guarantee the same initialization of models and the order of data samples. In this simple example, for each epoch, we did 20 times mini-batch training with batch size set as 64, thus in total 2000 training steps.

## B. Details of MA-TSD

### B.1. Transition matrix example

As depicted in Figure 6, we consider the situation of $L = 6$ as an naive example to illustrate how we get the square transition matrix $\boldsymbol{K}_t$ with moving average.

### B.2. Combination of denosing networks and conditioning networks

The combination of denosing networks and conditioning networks are shown in Figure 7. The Encoder embeds the noisy $\boldsymbol{z}_t$ and fuse with the position embedding of $t$. In the conditional generation, the condition $\boldsymbol{c}$ is also encoded and fused together. Then, the Decoder will output the prediction $\hat{\boldsymbol{z}}_0$, and the conditional decoder will output the $\hat{\sigma}, \hat{\mu}$ for denormalization, if needed. During inference, $\boldsymbol{z}_{t-1}$ will be obtained by $\boldsymbol{z}_t$ and $\hat{\boldsymbol{z}}_0$.

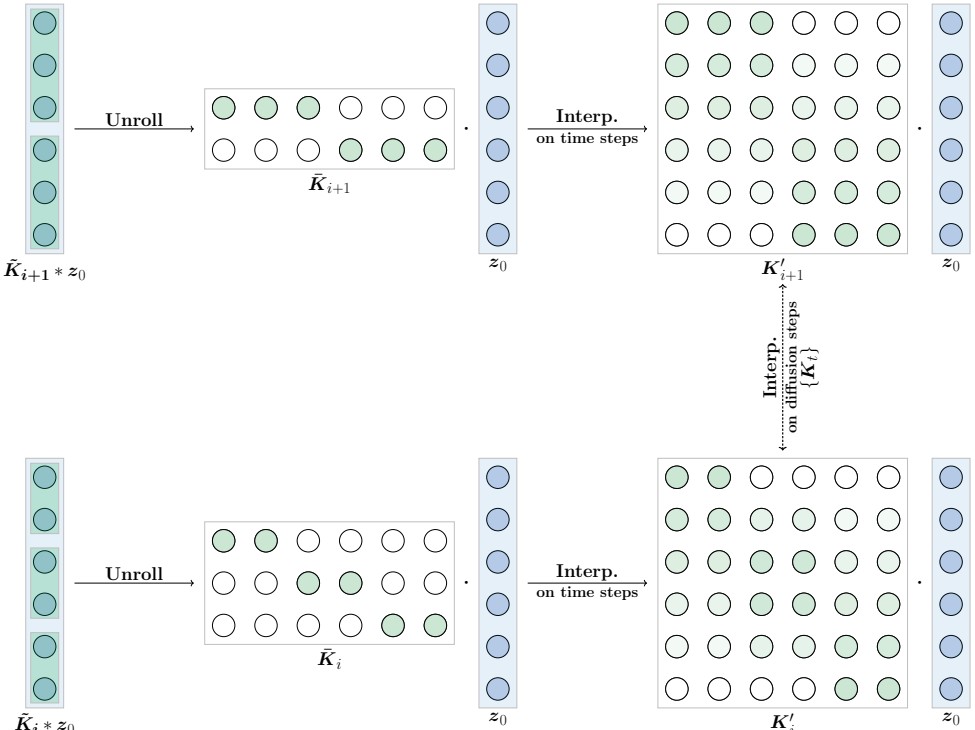

*Figure 6.* Example at $k_1 = 2$ and $k_2 = 3$. The convolution kernels are first unrolled to a matrix, and then interpolate along side the time steps to be square. Across the diffusion steps, the transition matrices are also interpolated.

### B.3. Proofs and Derivations

#### B.3.1. DECREASE RATIO $\gamma_t$ CALCULATION

We denote the decrease ratio calculated over the $z_0$ as $\gamma_t^z$:

$$\gamma_t^z = \mathbb{E}_{z_0 \sim q(z_0)} \left[ \frac{\sigma\left(K_t z_0\right)}{\sigma(z_0)} \right] = \mathbb{E}_{x_0 \sim q(x_0)} \left[ \frac{\sigma\left(K_t \frac{x_0 - \mu(x_0)}{\sigma(x_0)}\right)}{\sigma\left(\frac{x_0 - \mu(x_0)}{\sigma(x_0)}\right)} \right] = \mathbb{E}_{x_0 \sim q(x_0)} \left[ \frac{\sigma\left(K_t (x_0 - \mu(x_0))\right)}{\sigma(x_0 - \mu(x_0))} \right]$$

$$= \mathbb{E}_{x_0 \sim q(x_0)} \left[ \frac{\sigma\left(K_t x_0 - \mu(x_0)\right)}{\sigma(x_0 - \mu(x_0))} \right] = \mathbb{E}_{x_0 \sim q(x_0)} \left[ \frac{\sigma\left(K_t x_0\right)}{\sigma(x_0)} \right] = \gamma_t,$$

where the first equal in the second line is because moving average doesn't change the mean value of $x_0$, i.e. $K_t(x_0 - \mu(x_0)) = K_t x_0 - \mu(x_0)$, and the second equal in the second line is because $\sigma(x_0 + const.) = \sigma(x_0)$. Therefore, it's the same to calculate $\gamma_t$ over $z_0$ and $x_0$.

#### B.3.2. THE CHOICE OF $q(z_{t-1}|z_t, z_0)$

Given the defined $q(z_{1:T}|z_0) := q(z_T|z_0) \prod_{t=2}^{T} q(z_{t-1}|z_t, z_0)$, the defined $q(z_{t-1}|z_t, z_0)$ in Equation (14) and $q(z_T|z_0) := \mathcal{N}\left(z_T; K_T z_0, \beta_T^2 I\right)$, we can have $q(z_t|z_0) = \mathcal{N}\left(z_t; K_t z_0, \beta_t^2 I\right)$ for all $t$. The proof is similar to that of (Song et al., 2021a), with the transition generalized to $K_t x_0$

*Proof.* We can prove the above statement through an induction argument. Assume that for $t \leq T$, $q(z_t|z_0) = \mathcal{N}\left(z_t; K_t z_0, \beta_t^2 I\right)$ stands, and if $q(z_{t-1}|z_0) = \mathcal{N}\left(z_{t-1}; K_{t-1} z_0, \beta_{t-1}^2 I\right)$ also stands, then we can prove it from $t = T$ to $t = 1$ with the initial $q(z_T|z_0) := \mathcal{N}\left(z_T; K_T z_0, \beta_T^2 I\right)$.

First, via the marginalization, we have:

$$q(z_{t-1}|z_0) = \int_{x_t} q(z_{t-1}|z_t, z_0) q(z_t|z_0) dz_t. \tag{20}$$

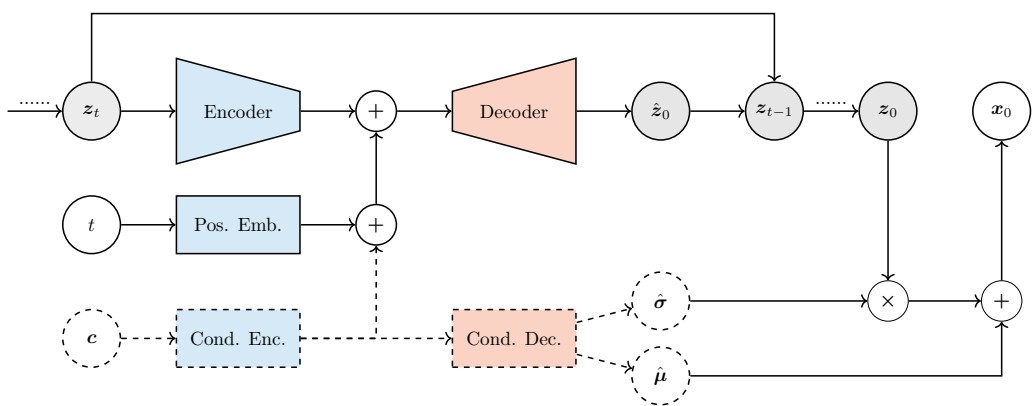

*Figure 7.* The denoising and statistics prediction models for time series forecasting. To include the unconditional generation $c = \varnothing$, we make the dashed line for illustration. Pos. Emb.=Position Embedder, Cond. Enc.=Condition Encoder, Cond. Dec.=Condition Decoder.

With the given $q(z_t|z_0) = \mathcal{N}\left(z_t; K_t z_0, \beta_t^2 I\right)$ and $q(z_{t-1}|z_t, z_0) = \mathcal{N}\left(K_{t-1} z_0 + \frac{\sqrt{\beta_{t-1}^2 - \eta_t^2}}{\beta_t}(z_t - K_t z_0), \eta_t^2 I\right)$, we can deduce that $q(z_{t-1}|z_0)$ is also Gaussian, with the mean $\mu_{z_{t-1}}$ and the variance $\Sigma_{z_{t-1}}$:

$$
\begin{aligned}
\mu_{z_{t-1}} &= K_{t-1} z_0 + \frac{\sqrt{\beta_{t-1}^2 - \eta_t^2}}{\beta_t}(K_t z_0 - K_t z_0) = K_{t-1} z_0, \\
\Sigma_{z_{t-1}} &= \left(\eta_{t-1}^2 + \beta_t^2 \cdot \frac{\beta_{t-1}^2 - \eta_t^2}{\beta_t^2}\right) I = \beta_{t-1}^2 I.
\end{aligned}
\tag{21}
$$

Therefore, $q(z_{t-1}|z_0) = \mathcal{N}(K_{t-1} z_0, \beta_{t-1}^2 I)$ holds and the inductive argument can be processed.

### B.3.3. BACKWARD ODE

Let us first consider our backward process without acceleration, given by:

$$
z_{t-1} = K_{t-1} f_\theta(z_t, t, c) + \frac{\sqrt{\beta_{t-1}^2 - \eta_t^2}}{\beta_t} (z_t - K_t f_\theta(z_t, t, c)) + \eta_t \epsilon.
\tag{22}
$$

When $\eta_t = 0$, we can have:

$$
z_{t-1} = K_{t-1} f_\theta(z_t, t, c) + \frac{\beta_{t-1}}{\beta_t} (z_t - K_t f_\theta(z_t, t, c))
\tag{23}
$$

After reformulation, we further have:

$$
\frac{z_{t-1}}{\beta_{t-1}} - \frac{z_t}{\beta_t} = \left(\frac{K_{t-1}}{\beta_{t-1}} - \frac{K_t}{\beta_t}\right) f_\theta(z_t, t, c),
\tag{24}
$$

which can be viewed as the numerical solution with Euler method to the following ODE with the discrete step $\Delta t = 1$:

$$
d\left(\frac{z_t}{\beta_t}\right) = f_\theta^\top(z_t, t) d\left(\frac{K_t^\top}{\beta_t}\right).
\tag{25}
$$

Now, we consider the even interpolation between $\{K_i'\}_i^n$ where we anchor the original $K_i'$, and linearly inject $m$ intermediate matrices between $K_i'$ and $K_{i+1}'$. Then the collected diffusion step subset $\{t_i^*\}_1^n$ are essentially $\{i(m+1)\}_1^n = \{m+1, 2(m+1), \cdots, n(m+1)\}$. Therefore, backward with $\{t_i^*\}_1^n$ in this situation is basically Euler method with the discrete step $\Delta t = m+1$.

### B.3.4. HYBRID OPTIMIZATION

The conditional diffusion loss function without instance normalization over $\boldsymbol{x}_0$ can be written by:

$$\mathcal{L}_{\boldsymbol{x}} = \mathbb{E}_{\boldsymbol{x}_0, \boldsymbol{c}, t} \left[ \| h_\Theta(\boldsymbol{x}_t, t, \boldsymbol{c}) - \boldsymbol{x}_0 \|_2^2 \right], \tag{26}$$

where we use $h_\Theta$ to distinguish from $f_\theta$ and $g_\omega$ mentioned previously.

Further, we can express the $\boldsymbol{x}_0$ into: $\boldsymbol{x}_0 = \boldsymbol{z}_0 \cdot \sigma(\boldsymbol{x}_0) + \mu(\boldsymbol{x}_0)$, and then parameterize the denosing network accordingly: $h_\Theta(\boldsymbol{x}_t, t, \boldsymbol{c}) = f_\theta(\boldsymbol{z}_t, t, \boldsymbol{c}) \cdot g_\omega^\sigma(\boldsymbol{c}) + g_\omega^\mu(\boldsymbol{c})$

For simplicity, we denote $l_{\boldsymbol{x}} = \| h_\Theta(\boldsymbol{x}_t, t, \boldsymbol{c}) - \boldsymbol{x}_0 \|_2$, and have:

$$
\begin{align}
l_{\boldsymbol{x}}^2 &= \| h_\Theta(\boldsymbol{x}_t, t, \boldsymbol{c}) - \boldsymbol{x}_0 \|_2^2 \tag{27} \\
&= \| f_\theta(\boldsymbol{z}_t, t, \boldsymbol{c}) \cdot g_\omega^\sigma(\boldsymbol{c}) + g_\omega^\mu(\boldsymbol{c}) - \boldsymbol{z}_0 \cdot \sigma(\boldsymbol{x}_0) - \mu(\boldsymbol{x}_0) \|_2^2 \tag{28} \\
&\leq \left( \| f_\theta(\boldsymbol{z}_t, t, \boldsymbol{c}) \cdot g_\omega^\sigma(\boldsymbol{c}) - \boldsymbol{z}_0 \cdot \sigma(\boldsymbol{x}_0) \|_2 + \| g_\omega^\mu(\boldsymbol{c}) - \mu(\boldsymbol{x}_0) \|_2 \right)^2 \quad \text{(Triangle inequality)} \tag{29} \\
&= \left( \| f_\theta(\boldsymbol{z}_t, t, \boldsymbol{c}) \cdot g_\omega^\sigma(\boldsymbol{c}) - \boldsymbol{z}_0 \cdot \sigma(\boldsymbol{x}_0) + \boldsymbol{z}_0 \cdot g_\omega^\sigma(\boldsymbol{c}) - \boldsymbol{z}_0 \cdot g_\omega^\sigma(\boldsymbol{c}) \|_2 + l_\mu \right)^2 \tag{30} \\
&= \left( \| \boldsymbol{z}_0 \cdot (g_\omega^\sigma(\boldsymbol{c}) - \sigma(\boldsymbol{x}_0)) + g_\omega^\sigma(\boldsymbol{c}) \cdot (f_\theta(\boldsymbol{z}_t, t, \boldsymbol{c}) - \boldsymbol{z}_0) \|_2 + l_\mu \right)^2 \tag{31} \\
&\leq \left( \| \boldsymbol{z}_0 \|_\infty \| g_\omega^\sigma(\boldsymbol{c}) - \sigma(\boldsymbol{x}_0) \|_2 + \| g_\omega^\sigma(\boldsymbol{c}) \|_2 \| (f_\theta(\boldsymbol{z}_t, t, \boldsymbol{c}) - \boldsymbol{z}_0) \|_2 + l_\mu \right)^2 \quad \text{(Triangle inequality)} \tag{32} \\
&= \left( \| \boldsymbol{z}_0 \|_\infty l_\sigma + \| g_\omega^\sigma(\boldsymbol{c}) \|_2 l_{\boldsymbol{z}} + l_\mu \right)^2 \tag{33} \\
&\leq \left( \| \boldsymbol{z}_0 \|_\infty^2 + \| g_\omega^\sigma(\boldsymbol{c}) \|_2^2 + 1 \right) \left( l_{\boldsymbol{z}}^2 + l_\mu^2 + l_\sigma^2 \right) \quad \text{(Cauchy inequality).} \tag{34}
\end{align}
$$

Further, we can scale $\| \boldsymbol{z}_0 \|_\infty$ to the maximum absolute value of the all the $\boldsymbol{z}_0$ over the dataset. Then, for $g_\omega^\sigma(\boldsymbol{c})$, it tries to approximate the real standard deviation, $g_\omega^\sigma(\boldsymbol{c}) \approx \sigma(\boldsymbol{x}_0)$, and the maximum $\sigma(\boldsymbol{x}_0)$ over the whole training dataset $q(\boldsymbol{x}_0)$ can be obtained. We can further limit the output of the $g_\omega^\sigma$ to the largest $\sigma(\boldsymbol{x}_0)$. Therefore, we can scale $\left( \| \boldsymbol{z}_0 \|_\infty^2 + \| g_\omega^\sigma(\boldsymbol{c}) \|_2^2 + 1 \right) \leq \left( \boldsymbol{z}_{\max} + \sigma_{\max} + 1 \right) = \lambda$

Therefore, we have:

$$
\begin{align}
\mathcal{L}_{\boldsymbol{x}} &= \mathbb{E}_{\boldsymbol{x}_0, \boldsymbol{c}, t} \left[ l_x^2 \right] \tag{35} \\
&\leq \mathbb{E}_{\boldsymbol{x}_0, \boldsymbol{c}, t} \left[ \left( \| \boldsymbol{z}_0 \|_\infty^2 + \| g_\omega^\sigma(\boldsymbol{c}) \|_2^2 + 1 \right) \left( l_{\boldsymbol{z}}^2 + l_\mu^2 + l_\sigma^2 \right) \right] \tag{36} \\
&\leq \mathbb{E}_{\boldsymbol{x}_0, \boldsymbol{c}, t} \left[ \lambda \left( l_{\boldsymbol{z}}^2 + l_\mu^2 + l_\sigma^2 \right) \right] \tag{37}
\end{align}
$$

We compare the hybrid loss function:

$$
\begin{align}
\mathcal{L}_{\text{hybrid}} &= \lambda_{\boldsymbol{z}} \mathcal{L}_{\boldsymbol{z}} + \lambda_\mu \mathcal{L}_\mu + \lambda_\sigma \mathcal{L}_\sigma \notag \\
&= \mathbb{E}_{\boldsymbol{x}_0, \boldsymbol{c}, t} \left[ \| \lambda_{\boldsymbol{z}} f_\theta(\boldsymbol{z}_t, t, \boldsymbol{c}) - \boldsymbol{z}_0 \|_2^2 + \lambda_\mu \| g_\omega^\mu(\boldsymbol{c}) - \mu(\boldsymbol{x}_0) \|_2^2 + \lambda_\sigma \| g_\omega^\sigma(\boldsymbol{c}) - \sigma(\boldsymbol{x}_0) \|_2^2 \right] \tag{38} \\
&= \mathbb{E}_{\boldsymbol{x}_0, \boldsymbol{c}, t} \left[ \lambda_{\boldsymbol{z}} l_{\boldsymbol{z}}^2 + \lambda_\mu l_\mu^2 + \lambda_\sigma l_\sigma^2 \right]. \notag
\end{align}
$$

We can see that if the $\lambda_{\boldsymbol{z}} = \lambda_\mu = \lambda_\sigma = \lambda$, then $\mathcal{L}_{\boldsymbol{x}} \leq \mathcal{L}_{\text{hybrid}}$.

### B.4. Differences from Blurring Diffusion Models (Hoogeboom & Salimans, 2023)

From a high-level perspective, BDM and ours shared a similar idea, i.e., building the degradation process with low-pass filters (blurring in BDM, MA in ours). However, there exist clear distinctions.

**Filtering space:** We filtered the data in the time space by convolution (matrix multiplication), $q(\boldsymbol{x}_t \mid \boldsymbol{x}_0) = \mathcal{N}(\boldsymbol{K}_t \boldsymbol{x}_0, \beta_t^2 \boldsymbol{I})$ while BDM blurs the images in the frequency domain and transform back to the pixel domain (using convolution theorem), i.e., $q(\boldsymbol{x}_t \mid \boldsymbol{x}_0) = \mathcal{N}(\boldsymbol{V} \boldsymbol{\alpha}_t \boldsymbol{V}^\top \boldsymbol{x}_0, \sigma_t^2 \boldsymbol{I})$, where $\boldsymbol{V}^\top, \boldsymbol{V}$ are Discrete Cosine Transform (DCT) and Inverse DCT (IDCT), and $\boldsymbol{\alpha}_t$ represent the frequency response of Gaussian blurring kernel, a diagonal matrix, whose each entry $\alpha_t^i \in (0, 1]$ is a coefficient of $i^{\text{th}}$ frequency component. For low pass filters, $\forall t$, $\alpha_t^i$ decreases by $i$ until (nearly) zero to suppress high frequency components.

**Markovian or not:** Since $\boldsymbol{\alpha}_t$ is diagonal, BDM proposed that for each frequency component, a standard Markovian DDPM can be constructed. The one-step transition is accordingly defined, i.e. $q(\boldsymbol{u}_t \mid \boldsymbol{u}_{t-1}) = \mathcal{N}(\boldsymbol{\alpha}_{t|t-1} \boldsymbol{u}_{t-1}, \sigma_{t|t-1}^2 \boldsymbol{I})$,

where $\boldsymbol{u}_t = \boldsymbol{V}^\top \boldsymbol{x}_0$ is the frequency representation and $\boldsymbol{\alpha}_{t|t-1} = \boldsymbol{\alpha}_t / \boldsymbol{\alpha}_{t-1}$. However, dividing $\boldsymbol{\alpha}_{t-1}$ could be numerically unstable in practice. As we mentioned above, $\alpha_t^i$ could become to be (nearly) zero for larger $i$, so dividing $\boldsymbol{\alpha}_{t-1}$ is unstable for all diffusion steps. Though some epsilons can be added to ensure stable division, tiny errors in the high frequency components will still be amplified a lot through the iterative backward process. Chances are that the generated data are dominated by the improperly amplified high frequency components. Therefore, when it comes to designing a diffusion process with non-reversible low-pass filters, we believe it's improper to follow the Markovian assumption and define the necessary $q(\boldsymbol{x}_t \mid \boldsymbol{x}_{t-1})$. Instead, in our framework, faced with similar non-reverible MA, we bypassed the definition of $q(\boldsymbol{x}_t \mid \boldsymbol{x}_{t-1})$, assumed $q(\boldsymbol{x}_{1:T} \mid \boldsymbol{x}_0)$ non-Markovian (in the DDIM-style) and then delicately defined $q(\boldsymbol{x}_{t-1} \mid \boldsymbol{x}_t, \boldsymbol{x}_0)$ to satisfy $q(\boldsymbol{x}_t \mid \boldsymbol{x}_0)$ for all t. Thus, whether it's Markovian or not is another distinct difference from ours and BDM.

**Noise schedule:** BDM designed $\boldsymbol{\alpha}_t = a_t \boldsymbol{d}_t$, where $\boldsymbol{d}_t$ is the frequency response of blurring kernel and $a_t \in [0, 1]$ is an extra scaler decreasing by $t$, and the noise schedule $\sigma_t = 1 - a_t^2$. In our framework, the noise schedule $\beta_t$ is dataset-based, chosen regarding the variance decrease caused by MA (Equation (13)) on different datasets.

In summary, despite the similar high-level idea, there exist clear differences between BDM and ours. BDM tried to fit in the standard Markovian DDPM framework, while we reformulated a framework with special adaptation on moving average filters and time series.

## C. Experiment details

We launch our experiments on a single NVIDIA GeForce RTX 4090 24GB GPU.

### C.1. Synthesis

We follow the setting of KoVAE (Naiman et al., 2024b) and ImagenTime (Naiman et al., 2024a) on three datasets, ETTh2, Exchange, and ECG (medical time series)[1], i.e. $L = 24$. KoVAE is a SOTA VAE-based time series generative model while ImagenTime is a diffuison-based one.

We also include discriminative score (Disc. Score) and predictive score (Pred. Score) as additional metrics for evaluating the fidelity and usefulness of synthetic time series, as the following table shows.

*Table 5.* Results on standard time series synthesis

| DATASET | MODEL | DISC. SCORE | PRED. SCORE | CONTEXT-FID |
|---------|-------|-------------|-------------|-------------|
| ETTH2 | KOVAE | 0.069 | 0.034 | 0.258 |
| | IMAGENTIME | 0.053 | 0.054 | 0.118 |
| | **OURS** | **0.044** | **0.026** | **0.075** |
| EXCHANGE | KOVAE | 0.137 | 0.038 | 1.520 |
| | IMAGENTIME | 0.129 | 0.067 | 1.112 |
| | **OURS** | **0.030** | **0.027** | **0.083** |
| ECG | KOVAE | 0.459 | 0.081 | 1.206 |
| | IMAGENTIME | 0.400 | 0.079 | 1.223 |
| | **OURS** | **0.345** | **0.076** | **0.979** |

We can see that compared to the SOTA time series generative models, our method still shows salient improvements in discriminative score and predictive score, illustrating the capability of generating high-fidelity and useful synthetic time series samples.

### C.2. Forecasting

The length of the look-back window is set as 96, and the target time series length is set as $L \in \{96, 192, 336, 720\}$. Such a setting is aligned with (Wang et al., 2024). Table 6 recorded the information of the forecasting datasets, Electricity[2], ETT[3],

---

[1] https://www.kaggle.com/datasets/devavratatripathy/ecg-dataset
[2] https://archive.ics.uci.edu/ml/datasets/ElectricityLoadDiagrams20112014
[3] https://github.com/zhouhaoyi/ETDataset

Table 6. Forecasting datasets

| Dataset | Resolution | Time steps | Description |
|---|---|---|---|
| Electricity | 1 hour | 26304 | Electricity consumptions |
| ETTh | 1 hour | 17420 | Oil temperature of power transformers |
| ETTm | 15 min | 69680 | Oil temperature of power transformers |
| Exchange | 1 day | 7588 | Panel data of exchange rates |
| Traffic | 1 hour | 17544 | Traffic loads |
| Weather | 10 min | 52695 | Meteorological indicators |

Exchange[4], Traffic[5] and Weather[6].

We set the batch size as 64, the learning rate as $2 \times 10^{-4}$, the training epoch as 100 with early stopping, and the diffusion step $T = 100$. To include the condition input, the condition encoder and decoder are all Multi-Layer Perceptrons (MLP). The hyperparameters of hybrid optimization are chosen as $\lambda_{\boldsymbol{z}} = \lambda_{\mu} = \lambda_{\sigma} = 1$. The specific network architectures can be referred to as our source code.

The diffusion models inferred for 100 times to calculate the metrics. For MSE, we averaged over 100 times to have deterministic forecasts, while for CRPS, we first calculated nine quantiles at $\{0.1, 0.2, \cdots, 0.9\}$, and then approximated CRPS.

For each target length, benchmarks and our proposed model are trained with 5 different random initialization seeds, and the full results on each target length are reported in Table 7.

**Non-diffusion time series forecasting benchmarks**. Although our paper focuses on how to improve the time series diffusion model, we also believe that it's necessary to include SOTA non-diffusion time series forecasting methods as a reference. Therefore, we included Autoformer (Wu et al., 2021), Non-stationary Transformer (Liu et al., 2022), and PatchTST (Nie et al., 2023) to compare deterministic forecasting performance. We run all models in the same setting mentioned above, i.e., $L = \{96, 192, 336, 720\}$, and the averaged MSEs over all $L$ is reported in Table 8.

Regarding overall performance, our model still ranks first among these benchmarks, though it is slightly inferior to ETTh2 and weather compared to PatchTST.

It should be noted that these SOTA architectures are particularly tailored for time series forecasting and well adapted to the benchmark datasets, while forecasting is one of the downstream applications of our proposed MA-TSD framework. Therefore, we think there exists great potential to accommodate the SOTA architectures into our MA-TSD framework to have a better forecasting performance in our future work.

### C.3. Super-resolution

For time series super-resolution, we set the length of the time series as $L = 576$, and also train with the batch size as 64, the learning rate as $2 \times 10^{-4}$, the training epoch as 100 with early stopping, the diffusion step $T = 100$.

The information of the datasets, MFRED (Meinrenken et al., 2020), Wind[7] and Solar[8] are listed in Table 9.

Table 9. Super-resolution datasets

| Dataset | Resolution | Time steps | Description |
|---|---|---|---|
| MFRED | 5 min | 25908 | Household electricity load in Manhattan |
| Wind | 5 min | 26496 | Power generation from Australian wind farms |
| Solar | 5 min | 52992 | Power generation from Australian PV panels |

---

[4] https://github.com/laiguokun/multivariate-time-series-data
[5] http://pems.dot.ca.gov
[6] https://www.bgc-jena.mpg.de/wetter/
[7] https://zenodo.org/records/4654858
[8] https://zenodo.org/records/8219786

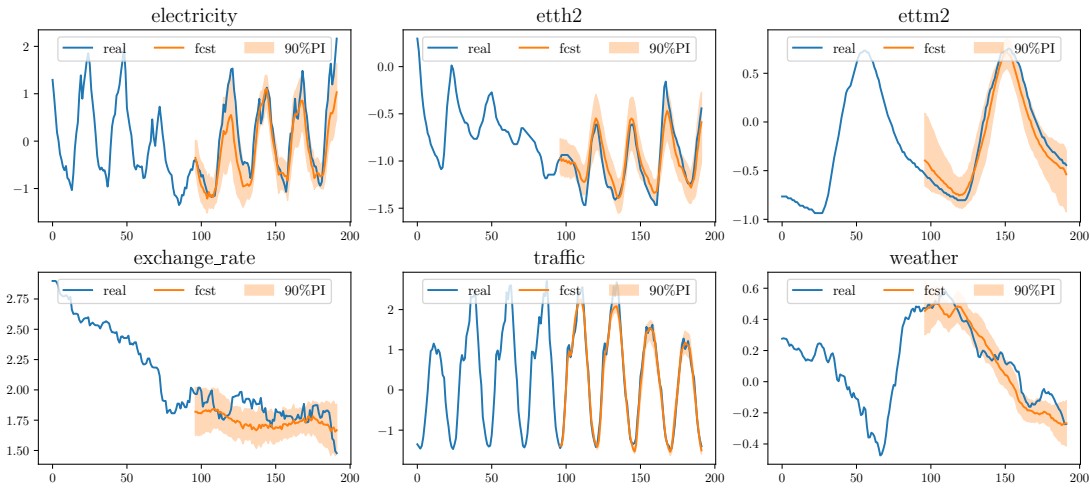

*Figure 8.* Forecasting samples on $L = 96$

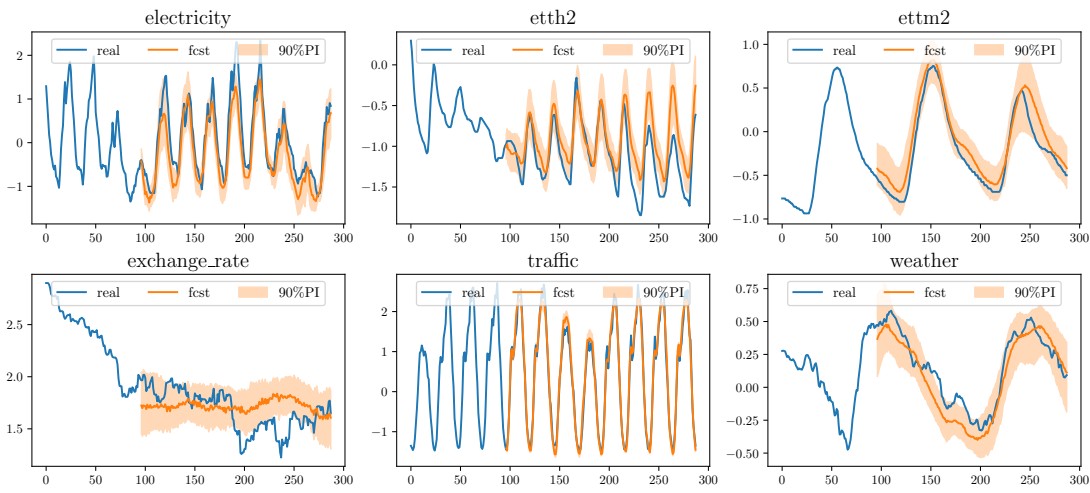

*Figure 9.* Forecasting samples on $L = 192$

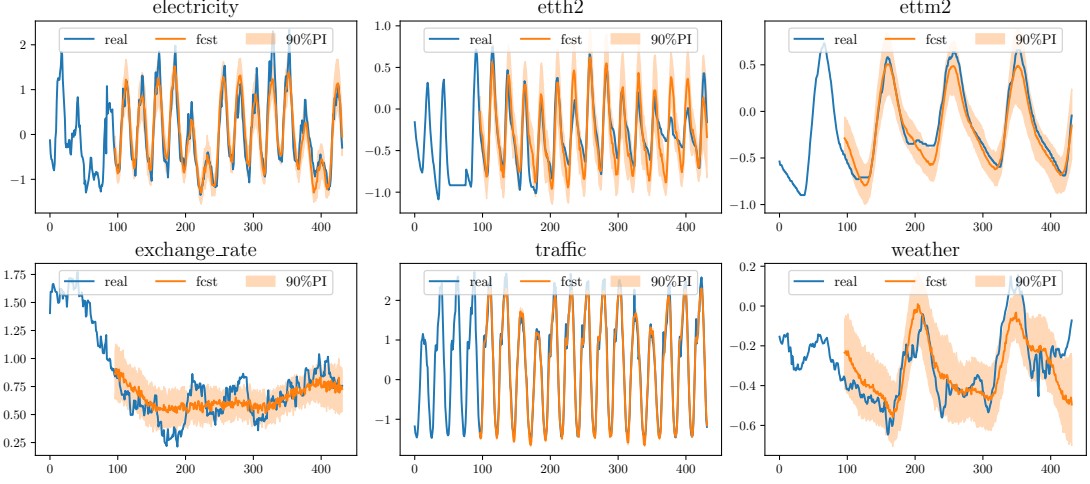

*Figure 10.* Forecasting samples on $L = 336$

*Table 7.* Forecasting performance measured in MSE and CRPS.

| Dataset | $L$ | MA-TSD | | mrdiff | | TMDM | | SSSD | | D3VAE | | CSDI | |
|---|---|---|---|---|---|---|---|---|---|---|---|---|---|
| | | MSE | CRPS | MSE | CRPS | MSE | CRPS | MSE | CRPS | MSE | CRPS | MSE | CRPS |
| Electricity | 96 | $\mathbf{0.288}_{\pm0.003}$ | $\mathbf{0.158}_{\pm0.001}$ | $0.496_{\pm0.008}$ | $0.226_{\pm0.004}$ | $\underline{0.348}_{\pm0.018}$ | $\underline{0.175}_{\pm0.006}$ | $0.976_{\pm0.043}$ | $0.309_{\pm0.004}$ | $0.816_{\pm0.078}$ | $0.306_{\pm0.014}$ | $0.358_{\pm0.022}$ | $0.168_{\pm0.006}$ |
| | 192 | $\mathbf{0.297}_{\pm0.006}$ | $\mathbf{0.160}_{\pm0.002}$ | $0.466_{\pm0.010}$ | $0.218_{\pm0.002}$ | $\underline{0.380}_{\pm0.020}$ | $\underline{0.182}_{\pm0.005}$ | $1.020_{\pm0.037}$ | $0.319_{\pm0.004}$ | $0.885_{\pm0.083}$ | $0.314_{\pm0.023}$ | $0.415_{\pm0.006}$ | $0.185_{\pm0.002}$ |
| | 336 | $\mathbf{0.347}_{\pm0.008}$ | $\mathbf{0.177}_{\pm0.003}$ | $0.496_{\pm0.009}$ | $0.224_{\pm0.006}$ | $\underline{0.423}_{\pm0.016}$ | $\underline{0.192}_{\pm0.002}$ | $1.035_{\pm0.040}$ | $0.324_{\pm0.006}$ | $0.808_{\pm0.126}$ | $0.307_{\pm0.021}$ | $0.459_{\pm0.011}$ | $0.195_{\pm0.003}$ |
| | 720 | $\mathbf{0.430}_{\pm0.018}$ | $\underline{0.204}_{\pm0.003}$ | $0.657_{\pm0.025}$ | $0.275_{\pm0.006}$ | $\underline{0.478}_{\pm0.025}$ | $\mathbf{0.203}_{\pm0.005}$ | $1.072_{\pm0.037}$ | $0.335_{\pm0.004}$ | $0.871_{\pm0.138}$ | $0.318_{\pm0.021}$ | $0.601_{\pm0.052}$ | $0.228_{\pm0.011}$ |
| ETTh2 | 96 | $\mathbf{0.136}_{\pm0.002}$ | $\mathbf{0.121}_{\pm0.001}$ | $\underline{0.149}_{\pm0.003}$ | $\underline{0.134}_{\pm0.001}$ | $0.208_{\pm0.015}$ | $0.141_{\pm0.006}$ | $0.590_{\pm0.039}$ | $0.27_{\pm0.011}$ | $1.577_{\pm0.721}$ | $0.464_{\pm0.106}$ | $0.212_{\pm0.033}$ | $0.140_{\pm0.011}$ |
| | 192 | $\mathbf{0.199}_{\pm0.007}$ | $\mathbf{0.149}_{\pm0.002}$ | $\underline{0.194}_{\pm0.004}$ | $\underline{0.155}_{\pm0.002}$ | $0.241_{\pm0.010}$ | $0.156_{\pm0.004}$ | $0.712_{\pm0.084}$ | $0.304_{\pm0.023}$ | $1.635_{\pm0.339}$ | $0.461_{\pm0.042}$ | $0.236_{\pm0.009}$ | $0.151_{\pm0.002}$ |
| | 336 | $\mathbf{0.229}_{\pm0.009}$ | $\mathbf{0.166}_{\pm0.002}$ | $\underline{0.236}_{\pm0.003}$ | $0.176_{\pm0.002}$ | $0.277_{\pm0.009}$ | $0.170_{\pm0.002}$ | $0.799_{\pm0.088}$ | $0.339_{\pm0.027}$ | $1.522_{\pm0.722}$ | $0.414_{\pm0.089}$ | $0.262_{\pm0.009}$ | $\underline{0.165}_{\pm0.006}$ |
| | 720 | $\underline{0.284}_{\pm0.017}$ | $\underline{0.191}_{\pm0.006}$ | $0.289_{\pm0.005}$ | $0.194_{\pm0.003}$ | $\mathbf{0.278}_{\pm0.006}$ | $\mathbf{0.169}_{\pm0.003}$ | $0.779_{\pm0.088}$ | $0.339_{\pm0.027}$ | $0.851_{\pm0.200}$ | $0.330_{\pm0.040}$ | $0.319_{\pm0.040}$ | $0.199_{\pm0.014}$ |
| ETTm2 | 96 | $\mathbf{0.070}_{\pm0.002}$ | $\mathbf{0.085}_{\pm0.001}$ | $0.122_{\pm0.042}$ | $0.105_{\pm0.015}$ | $\underline{0.090}_{\pm0.008}$ | $\underline{0.089}_{\pm0.004}$ | $0.549_{\pm0.061}$ | $0.264_{\pm0.017}$ | $3.743_{\pm0.637}$ | $0.719_{\pm0.059}$ | $1.609_{\pm0.417}$ | $0.427_{\pm0.067}$ |
| | 192 | $\mathbf{0.105}_{\pm0.003}$ | $\mathbf{0.105}_{\pm0.002}$ | $0.125_{\pm0.010}$ | $0.112_{\pm0.004}$ | $\underline{0.142}_{\pm0.008}$ | $0.114_{\pm0.004}$ | $0.782_{\pm0.070}$ | $0.332_{\pm0.018}$ | $2.934_{\pm0.639}$ | $0.624_{\pm0.077}$ | $1.763_{\pm0.389}$ | $0.438_{\pm0.054}$ |
| | 336 | $\mathbf{0.136}_{\pm0.004}$ | $\mathbf{0.121}_{\pm0.002}$ | $0.198_{\pm0.013}$ | $0.144_{\pm0.008}$ | $\underline{0.187}_{\pm0.020}$ | $\underline{0.132}_{\pm0.006}$ | $1.038_{\pm0.124}$ | $0.394_{\pm0.029}$ | $3.490_{\pm0.866}$ | $0.671_{\pm0.090}$ | $2.751_{\pm0.747}$ | $0.556_{\pm0.113}$ |
| | 720 | $\mathbf{0.185}_{\pm0.003}$ | $\mathbf{0.143}_{\pm0.002}$ | $0.235_{\pm0.023}$ | $0.157_{\pm0.007}$ | $\underline{0.297}_{\pm0.061}$ | $0.166_{\pm0.015}$ | $1.206_{\pm0.085}$ | $0.437_{\pm0.021}$ | $3.212_{\pm1.440}$ | $0.585_{\pm0.115}$ | $2.369_{\pm0.256}$ | $0.466_{\pm0.021}$ |
| Exchange | 96 | $\mathbf{0.098}_{\pm0.006}$ | $\underline{0.110}_{\pm0.004}$ | $\underline{0.102}_{\pm0.005}$ | $\mathbf{0.104}_{\pm0.003}$ | $0.392_{\pm0.096}$ | $0.196_{\pm0.025}$ | $2.572_{\pm0.200}$ | $0.620_{\pm0.022}$ | $0.674_{\pm0.131}$ | $0.290_{\pm0.037}$ | $0.170_{\pm0.083}$ | $0.123_{\pm0.021}$ |
| | 192 | $\mathbf{0.187}_{\pm0.005}$ | $\mathbf{0.158}_{\pm0.002}$ | $\underline{0.251}_{\pm0.011}$ | $\underline{0.170}_{\pm0.003}$ | $0.670_{\pm0.097}$ | $0.297_{\pm0.019}$ | $3.672_{\pm0.318}$ | $0.775_{\pm0.040}$ | $2.608_{\pm0.859}$ | $0.601_{\pm0.113}$ | $0.259_{\pm0.089}$ | $0.170_{\pm0.018}$ |
| | 336 | $\mathbf{0.333}_{\pm0.017}$ | $\underline{0.221}_{\pm0.005}$ | $\underline{0.456}_{\pm0.034}$ | $\mathbf{0.220}_{\pm0.006}$ | $0.929_{\pm0.116}$ | $0.333_{\pm0.011}$ | $3.131_{\pm0.088}$ | $0.710_{\pm0.017}$ | $2.851_{\pm0.495}$ | $0.674_{\pm0.063}$ | $0.699_{\pm0.316}$ | $0.302_{\pm0.037}$ |
| | 720 | $\mathbf{0.869}_{\pm0.140}$ | $\underline{0.382}_{\pm0.037}$ | $\underline{1.111}_{\pm0.069}$ | $\mathbf{0.353}_{\pm0.015}$ | $1.163_{\pm0.270}$ | $0.358_{\pm0.036}$ | $2.226_{\pm0.143}$ | $0.592_{\pm0.025}$ | $2.301_{\pm0.732}$ | $0.587_{\pm0.256}$ | $3.895_{\pm3.406}$ | $0.617_{\pm0.313}$ |
| Traffic | 96 | $\mathbf{0.166}_{\pm0.004}$ | $\mathbf{0.112}_{\pm0.003}$ | $0.269_{\pm0.002}$ | $0.153_{\pm0.002}$ | $\underline{0.209}_{\pm0.019}$ | $\underline{0.114}_{\pm0.004}$ | $1.943_{\pm0.017}$ | $0.453_{\pm0.002}$ | $4.714_{\pm2.358}$ | $0.729_{\pm0.149}$ | $0.278_{\pm0.017}$ | $0.145_{\pm0.006}$ |
| | 192 | $\mathbf{0.157}_{\pm0.003}$ | $\underline{0.109}_{\pm0.003}$ | $0.228_{\pm0.001}$ | $0.136_{\pm0.001}$ | $\underline{0.172}_{\pm0.010}$ | $\mathbf{0.105}_{\pm0.005}$ | $1.959_{\pm0.006}$ | $0.455_{\pm0.002}$ | $7.353_{\pm0.692}$ | $0.918_{\pm0.049}$ | $0.276_{\pm0.022}$ | $0.143_{\pm0.009}$ |
| | 336 | $\mathbf{0.157}_{\pm0.005}$ | $\underline{0.114}_{\pm0.004}$ | $0.225_{\pm0.011}$ | $0.137_{\pm0.007}$ | $\underline{0.161}_{\pm0.010}$ | $\mathbf{0.102}_{\pm0.005}$ | $1.965_{\pm0.015}$ | $0.458_{\pm0.003}$ | $5.155_{\pm2.550}$ | $0.734_{\pm0.153}$ | $0.310_{\pm0.034}$ | $0.153_{\pm0.010}$ |
| | 720 | $\underline{0.184}_{\pm0.010}$ | $\underline{0.128}_{\pm0.007}$ | $0.267_{\pm0.01}$ | $0.155_{\pm0.005}$ | $\mathbf{0.180}_{\pm0.010}$ | $\mathbf{0.109}_{\pm0.003}$ | $1.998_{\pm0.022}$ | $0.465_{\pm0.004}$ | $8.212_{\pm0.999}$ | $0.944_{\pm0.054}$ | $1.132_{\pm0.629}$ | $0.312_{\pm0.106}$ |
| Weather | 96 | $0.096_{\pm0.001}$ | $0.102_{\pm0.002}$ | $0.108_{\pm0.006}$ | $0.109_{\pm0.003}$ | $0.109_{\pm0.011}$ | $0.100_{\pm0.006}$ | $0.511_{\pm0.029}$ | $0.305_{\pm0.015}$ | $1.447_{\pm0.138}$ | $0.448_{\pm0.021}$ | $\mathbf{0.095}_{\pm0.002}$ | $\mathbf{0.089}_{\pm0.001}$ |
| | 192 | $\underline{0.146}_{\pm0.004}$ | $0.126_{\pm0.002}$ | $0.152_{\pm0.003}$ | $0.130_{\pm0.002}$ | $0.163_{\pm0.017}$ | $0.120_{\pm0.006}$ | $0.660_{\pm0.049}$ | $0.305_{\pm0.015}$ | $1.901_{\pm0.324}$ | $0.508_{\pm0.034}$ | $\mathbf{0.138}_{\pm0.004}$ | $\mathbf{0.109}_{\pm0.002}$ |
| | 336 | $0.225_{\pm0.006}$ | $0.161_{\pm0.005}$ | $\underline{0.221}_{\pm0.006}$ | $0.155_{\pm0.002}$ | $0.250_{\pm0.022}$ | $0.149_{\pm0.007}$ | $0.759_{\pm0.059}$ | $0.331_{\pm0.015}$ | $1.600_{\pm0.313}$ | $0.457_{\pm0.039}$ | $\mathbf{0.204}_{\pm0.004}$ | $\mathbf{0.133}_{\pm0.001}$ |
| | 720 | $0.362_{\pm0.007}$ | $0.211_{\pm0.003}$ | $\underline{0.350}_{\pm0.004}$ | $0.208_{\pm0.002}$ | $0.363_{\pm0.032}$ | $\underline{0.183}_{\pm0.008}$ | $0.833_{\pm0.062}$ | $0.351_{\pm0.016}$ | $1.237_{\pm0.143}$ | $0.386_{\pm0.023}$ | $\mathbf{0.338}_{\pm0.023}$ | $\mathbf{0.173}_{\pm0.006}$ |

*Table 8.* Average MSEs over prediction lengths $L = \{96, 192, 336, 720\}$.

| method | Electricity | ETTh2 | ETTm2 | exchange | traffic | weather | rank |
|---|---|---|---|---|---|---|---|
| Autoformer | 0.594 | 0.218 | 0.168 | 0.601 | 0.267 | 0.293 | 3.83 |
| Nonstationary Transformer | 0.367 | 0.230 | 0.146 | 0.440 | 0.229 | 0.278 | 2.83 |
| PatchTST | 0.412 | 0.202 | 0.122 | 0.500 | 0.179 | 0.189 | 1.83 |
| MA-TSD | 0.340 | 0.212 | 0.124 | 0.372 | 0.166 | 0.207 | **1.50** |

For MFRED and Solar, the original resolution is 10 seconds and 1 minute respectively, we resampled them to 5 minutes for alignment.

We unify the denoising networks of both our proposed MA-TSD and the DDPM as the DiT(Peebles & Xie, 2023). For consistency, we calculated as the MSE between the low-resolution inputs and the down-scaled super-resolution outputs. For Context-FID, we trained Autoencoders for each training dataset individually, obtained the time series embeddings of super-resolution outputs and the real high-resolution data, and calculated the Fréchet distance over these two embeddings.

The comparison of inference steps on each scale is shown in Figure 12. Here, though we conduct the experiments only on $\{3, 6, 12\}$ scales, we can theocratically calculate the expected inference steps on other scales according to Equation (17).

Visualization of super-resolution can be found in Figure 13, Figure 14 and Figure 15.

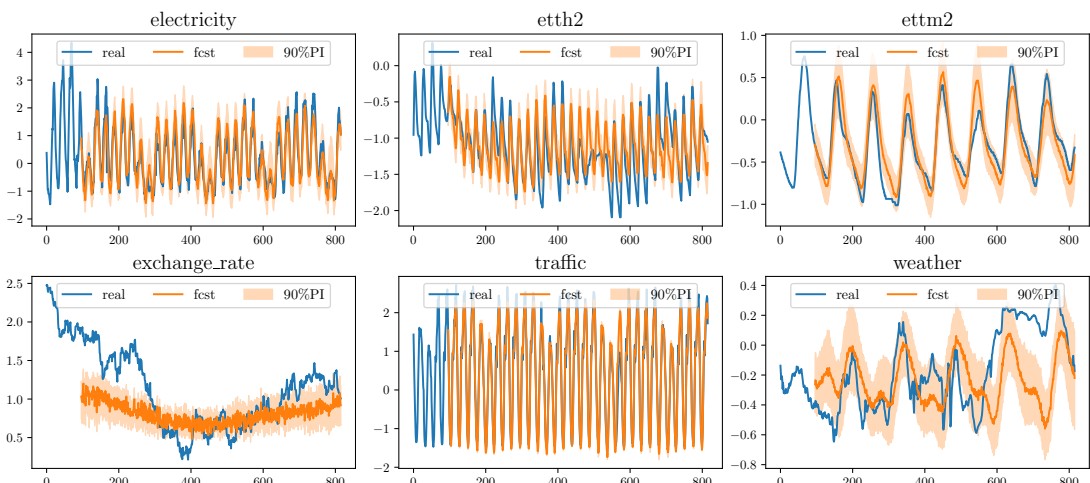

*Figure 11.* Forecasting samples on $L = 720$

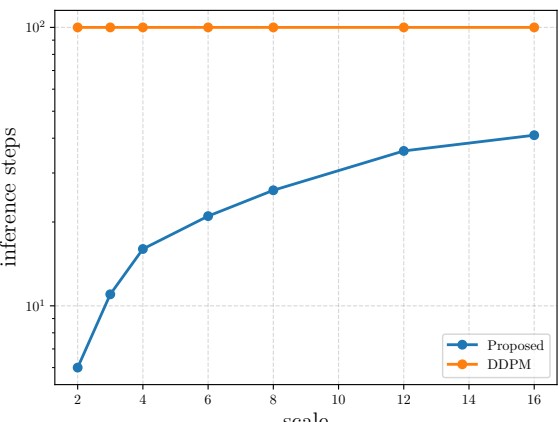

*Figure 12.* Comparison of inference steps under different super-resolution scales.

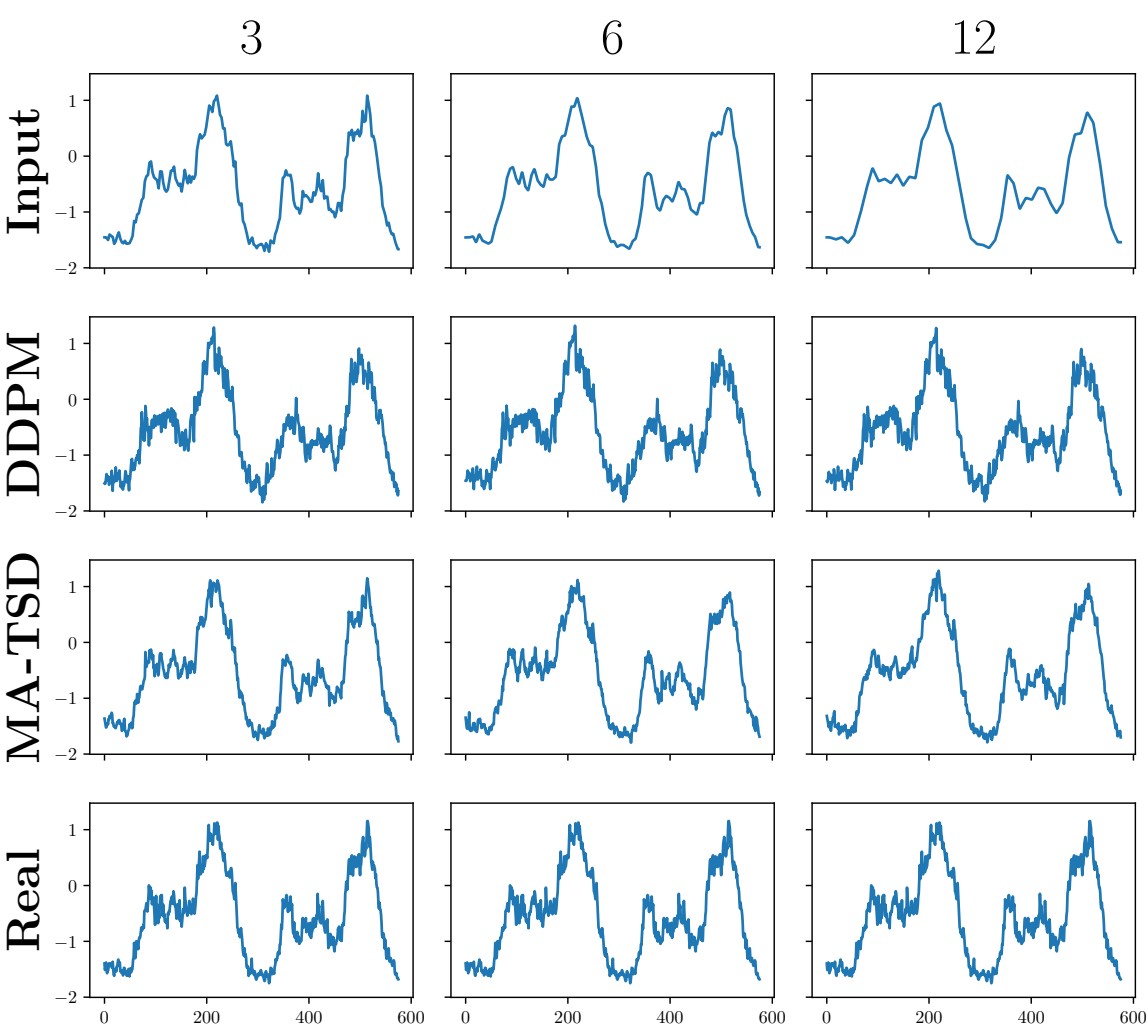

*Figure 13.* Super-resolution on MFRED dataset

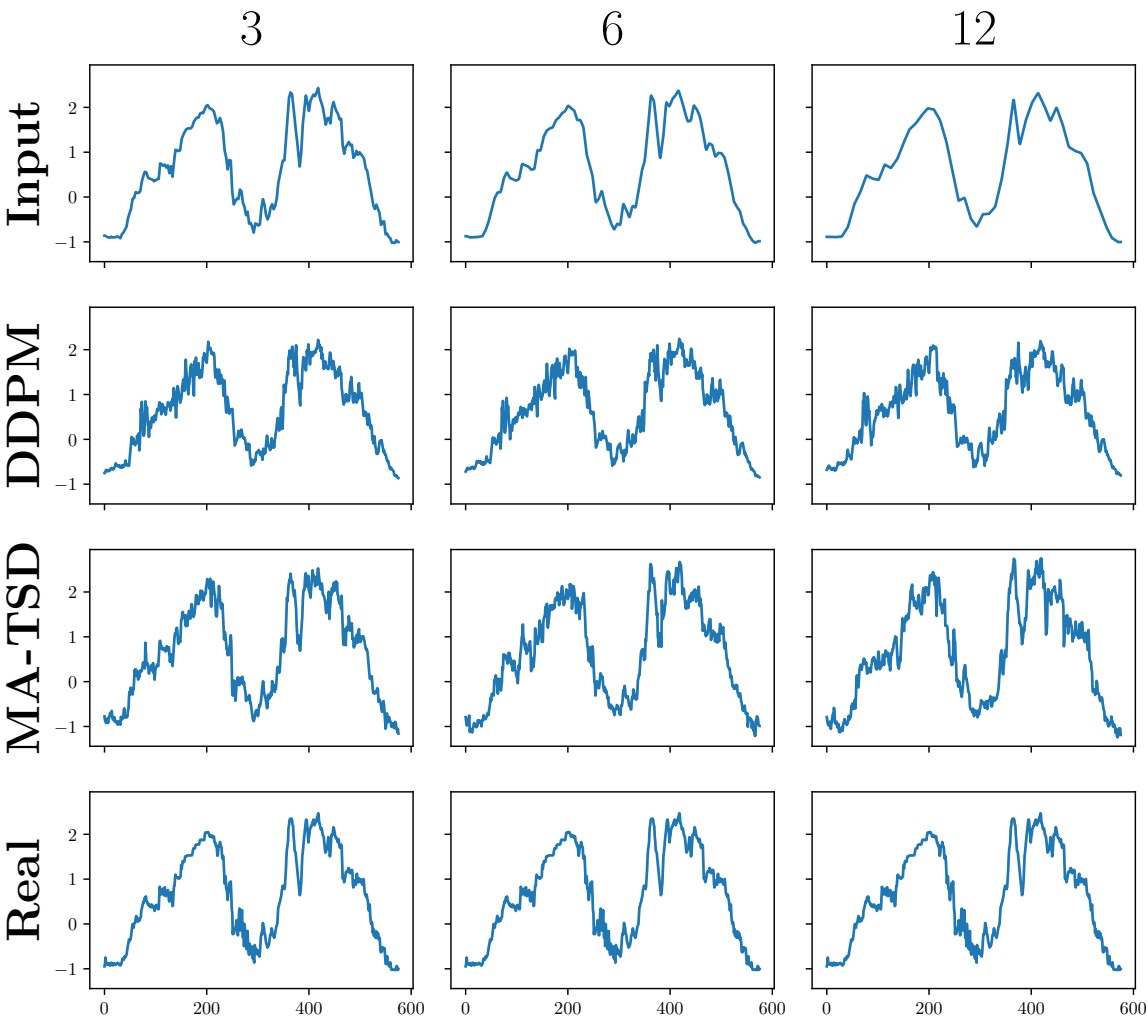

*Figure 14.* Super-resolution on Wind dataset

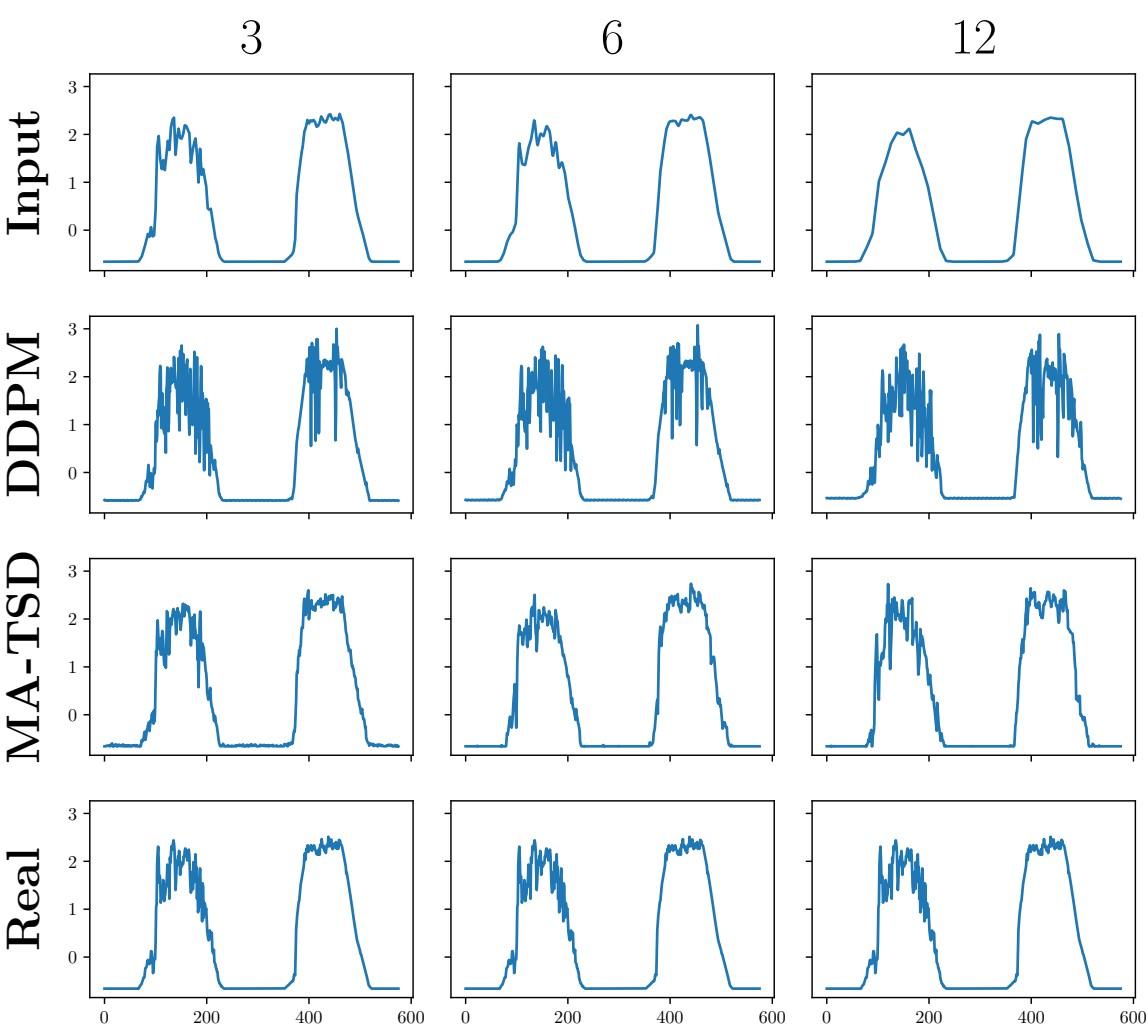

*Figure 15.* Super-resolution on Solar dataset

