# OpenReview forum: "A Non-isotropic Time Series Diffusion Model with Moving Average Transitions"
_ICML.cc/2025/Conference — ICML 2025 poster_

### Official Review · Reviewer_7aos · 2025-03-10

**Overall Recommendation:** 4

**Summary:**

This paper proposes a non-isotropic time series diffusion model with moving average transitions (MA-TSD). First, the authors empirically found that, when directly applying DDPM to time series data, the directions of model gradients at different diffusion steps are contradicted during training, which leads to unstable optimization. Second, they propose the moving average transition, which functions like a low-pass filter and keeps the informative low-frequency components. Meanwhile, they introduce the Non-isotropic forward process and the corresponding loss function, as well as the acceleration strategy (like DDIM) in inference. Extensive experiments are conducted, which demonstrate the salient performances of MA-TSD over existing DDPM-based diffusion models on time series-related tasks.

**Claims And Evidence:**

Yes, they are.

1. **Claim:** When directly applying DDPM to time series data, the directions of model gradients at different diffusion steps are contradicted during training. **Evidence:** Supported by Figure 2
2. **Claim:** The moving average schedule and the instance normalization are useful.  **Evidence:** Tables 1,2,4.
3. **Claim:** The backward process of MA-TSD can be naturally considered as time series super-resolution. **Evidence:** Table 3

**Essential References Not Discussed:**

No

**Experimental Designs Or Analyses:**

- In Figure 4, the factor-only accelerating strategy is only compared against randomly select subset strategy. To better validate its effectiveness, the authors should further include a comparison with uniformly sample strategy (e.g., the default DDIM approach, which uses uniformly spaced timesteps for acceleration).

other experimental designs and analyses are sound.

**Methods And Evaluation Criteria:**

Yes. They make sense.

**Other Comments Or Suggestions:**

A suggestion: The authors are suppose to ​cite relevant references in the tables (e.g., in table 1, table 2) to allow readers to easily trace the sources of compared methods.

**Other Strengths And Weaknesses:**

Strength:

- The empirical findings that direct application of standard diffusion to time series may cause gradient contradiction during training (due to the
  rapid decrease of low-frequency information in the diffusion process) is valuable，​and is insightful for the video generation community.
- The proposed ​moving average schedule and ​instance normalization demonstrate effectiveness.
- The ​factor-only acceleration strategy is effective.
- The paper provides ​thorough experimental validation and ablation studies.

Weakness

- The paper should include an ​ablation study on the parameter L， and analyze how varying L impacts the severity of gradient contradiction when directly applying DDPM to time series.
- The paper does not specify the ​model architecture (e.g., is it a 1D U-Net?)

**Questions For Authors:**

Beyond time-series forecasting, how applicable is this method to other scenarios, such as video generation/video prediction (treating frames as sequences)?

According to Figure 2, existing video diffusion models may also suffer from gradient contradiction. However, considering that existing video diffusion models typically predict shorter video clips, to what extent is this issue critically impactful? For example, How does the similarity maps in Figure 2 change when the $L$ varies?

**Relation To Broader Scientific Literature:**

The empirical findings that directly applying DDPM on time series data will cause contradicted gradients might be helpful for the long video generation task.

**Theoretical Claims:**

This paper does not have any theoretical claims.

---

> ### Author Rebuttal · Authors · 2025-03-31
>
> We are grateful for your valuable suggestions and insightful comments. We'd like to reply as follows:
>
> **1. Uniform sampling strategy**
>
> We've added the default DDIM sampling strategy to our experiments. Specifically, for a given sampling budget, we start with t=T and uniformly select the sampling steps. The results can be seen at (https://anonymous.4open.science/r/ICML2025_rebuttal-134C/sample_strategy.png). We can see that uniform sampling did offer a better result than random sampling, but the marginal gain on the performance decreases by $\tau/T$. Compared to both random and uniform strategies, our factor-only strategy also shows superior performance.
>
>
> **2. Ablation on $L$**
>
> We did an ablation study on the sequence length $L={48, 96, 192}$ and recorded the gradient similarity in this link (https://anonymous.4open.science/r/ICML2025_rebuttal-134C/ablation_L.png). We observe that for all L, gradient conflicts still exist when directly applying DDPM on time series data, and our method can keep alleviating. It should be noted that the severity of such conflicts is not changed with L varying. Instead, we think the gradient conflicts mainly come from the degradation design, and in our framework, MA always keeps more low-frequency information across different diffusion steps. These low-frequency components could serve as common information during diffusion training. Thus, the inputs for denoising models are more structurally similar, and consequently, the gradients across different steps are similar.
>
>
> **3. Backbone architecture**
>
> For the denoising backbone, we mainly rely on the DiT, with an adaptation to 1D time series. For the condition encoder in the forecasting task, we adopt an MLP with RevIN[1], which is widely used in TS forecasting. In our paper, we don't focus much on the neural architecture design, and we believe that the backbones can be replaced with other reasonable networks. Combining the SOTA time series architecture with our framework could also be one of our future works.
>
> **4. Applicable to video generation**
>
> We appreciate your insightful comments for inspiring us to explore more of our framework. Though we are not experts in the field of video generation, we still try to evaluate our framework on a simple "video-like" time series.
>
> Specifically, we believe that the transition between frames in a video should be smooth and mild because even the most common frame rate (like 24 FPS) is higher than the sampling rate of time series data. Thus, we manually create a pure 2-D signal: $x(t)=\sqrt{0.5 t} + 0.7 \cos(t) + 0.3 \sin(4 t), y(t)=\sqrt{0.2 t} + 0.8 \cos(t) + 0.2 \sin(3 t)$ with $t\in[0, 4\pi]$ recorded with 1280 time steps (mimicking a 53s video in 24FPS), and concatenate them as a bivariate time series. Then, we slide temporal windows with $L={48, 96, 192}$ and record the gradient similarity of directly using DDPM and our framework. As shown in (https://anonymous.4open.science/r/ICML2025_rebuttal-134C/simple_syn.png), we found a similar phenomenon, and our framework also works to ease the gradient conflicts to some extent. However, we think more framework adaptation is needed to bridge the large modality gap for video generation.
>
> In summary, our framework shows similar effects on our synthetic "video-like" time series, and we reckon that our high-level idea could also be applied to video generation with some necessary adaptation. We sincerely hope that our framework could also be enlightening and supportive for other fields beyond time series.
>
>
> [1] Reversible Instance Normalization for Accurate TSF Against Distribution Shift

---

> > ### Comment · Reviewer_7aos · 2025-04-02
> >
> > Thank you for your reply.
> >
> > I sincerely appreciate the additional experiments, especially the use of 2D time-series data to mimic video signals. All the results support the claims in the paper.
> >
> > Although the authors are not experts in video generation, they have made a commendable effort to demonstrate the promising applicability of their method to this field.
> >
> >
> > A minor suggestion: Currently, the "uniform sampling" curve does not include a confidence interval like that of "random sampling". This might be due to the tight rebuttal schedule, which constrains the repeated experiment run. It would be beneficial to include the confidence interval in a future version.
> >
> >
> > All my concerns are solved, and I will raise my rating to accept.

---

> > > ### Author Response · Authors · 2025-04-02
> > >
> > > We really appreciate your acknowledgment of our efforts, as well as all of your kind comments and advice that help us to improve our work.
> > >
> > > We are also grateful for understanding on the tight rebuttal schedule. Regarding your minor suggestion on the uniform sampling, we will add it in our future updated version.

---

### Official Review · Reviewer_EJkG · 2025-03-11

**Overall Recommendation:** 3

**Summary:**

This paper proposes MA-TSD (Moving Average Time Series Diffusion), a novel time series diffusion model that replaces the standard isotropic diffusion process with a moving average transition. The key motivation is that existing isotropic diffusion models degrade low and high-frequency components identically, which is problematic for time series data where low-frequency components often carry more essential information.

**Claims And Evidence:**

Yes

**Essential References Not Discussed:**

Yes, the paper is about a generative model, yet it does not address time series generation literature.

**Experimental Designs Or Analyses:**

No

**Methods And Evaluation Criteria:**

Yes, but a generative evaluation is missing, as described in the weaknesses response.

**Other Comments Or Suggestions:**

No

**Other Strengths And Weaknesses:**

**Strengths:**
- The use of different diffusion processes for different frequency levels is a novel and insightful approach.
- The gradient analysis provides valuable insights and strengthens the motivation behind the method.

**Weaknesses:**
- Despite proposing a new generative approach, the paper does not evaluate it on standard generative modeling benchmarks [1, 2, 3].
- In the super-resolution section, the improvement is demonstrated against only one diffusion model, limiting the assessment of its adaptability to other diffusion frameworks like EDM [4], FM [5], or alternative architectures.
- The forecasting benchmark lacks comparisons with state-of-the-art forecasting models. While I acknowledge the distinction between deterministic and probabilistic forecasting, a direct comparison remains essential.

[1] Interpretable Diffusion for General Time Series Generation

[2] Utilizing Image Transforms and Diffusion Models for
Generative Modeling of Short and Long Time Series

[3] Generative Modeling of Regular and Irregular Time Series Data via Koopman VAEs

[4] Elucidating the Design Space of Diffusion-Based Generative Models (EDM)

[5] Flow Matching for Generative Modeling

**Questions For Authors:**

See weaknesses

**Relation To Broader Scientific Literature:**

Time series generative models

**Theoretical Claims:**

No

---

> ### Author Rebuttal · Authors · 2025-03-31
>
> Thank you for the valuable feedback and the opportunity to strengthen our experiments. Below, we address each of the weaknesses:
>
> **1. Standard Time Series Generation**
>
> We agreed that the standard generation task is necessary for evaluating our model. Thus, we follow the setting of your mentioned benchmarks and experiment on three datasets, ETTh2, Exchange, and ECG (newly added, medical time series). We also include discriminative score and predictive score as additional metrics for evaluating the fidelity and usefulness, as the following table shows:
>
> | Dataset  |   Model    | Disc. Score | Pred. Score | Context-FID |
> | :------: | :--------: | :---------: | :---------: | :---------: |
> |  ETTh2   |   KoVAE    |    0.069    |    0.034    |    0.258    |
> |    | ImagenTime |    0.053    |    0.054    |    0.118    |
> |    |  **ours**  |  **0.044**  |  **0.026**  |  **0.075**  |
> | Exchange |   KoVAE    |    0.137    |    0.038    |    1.520    |
> | | ImagenTime |    0.129    |    0.067    |    1.112    |
> | |  **ours**  |  **0.030**  |  **0.027**  |  **0.083**  |
> | ECG |   KoVAE    |    0.459    |    0.081    |    1.206    |
> | | ImagenTime |    0.400    |    0.079    |    1.223    |
> | |  **ours**  |  **0.345**  |  **0.076**  |  **0.979**  |
>
>
> We can see that compared to the SOTA time series generation models, our method still shows salient improvements in discriminative score and predictive score, illustrating the capability of generating high-fidelity and useful synthetic time series samples.
>
> **2. Super-resolution(SR)**
>
> We included flow matching with Variance Preserving path (FM-VP) as our additional SR benchmark. The backbone used in FM-VP is kept the same as our original benchmark and model. The following table records the comparison:
>
> Table.1 Consistency error comparison
> | Scale | Model |   MFRED   |   Wind    |   Solar   |
> | :---: | :---: | :-------: | :-------: | :-------: |
> |   3   | ours  | **0.003** | **0.007** | **0.011** |
> |   3   | FM-VP |   0.033   |   0.064   |   0.052   |
> |   6   | ours  | **0.004** | **0.010** | **0.012** |
> |   6   | FM-VP |   0.024   |   0.050   |   0.036   |
> |  12   | ours  | **0.005** | **0.014** | **0.013** |
> |  12   | FM-VP |   0.016   |   0.035   |   0.024   |
>
> Table.2 Context-FID comparison
> | Scale | Model |   MFRED   |   Wind    |   Solar   |
> | :---: | :---: | :-------: | :-------: | :-------: |
> |   3   | ours  | **0.105** | **0.286** | **0.349** |
> |   3   | FM-VP |   1.348   |   4.231   |  0.795   |
> |   6   | ours  | **0.124** | **1.024** |   0.697   |
> |   6   | FM-VP |   1.338   |   4.468   | **0.685** |
> |  12   | ours  | **0.436** | **3.057** |   1.413   |
> |  12   | FM-VP |   1.522   |   4.874   | **0.862** |
>
> Despite the slight inferiority in Context-FID on the Solar dataset when the SR scale goes up, our proposed method still outperformed the FM-VP model generally, especially in terms of consistency. It should be noted again that we perform SR naturally through our backward process instead of retraining the whole conditional model like benchmarks. Therefore, we think our method provides a trade-off of (1) training overheads, (2) SR quality, and (3) consistency to the low-resolution input.
>
> **3. Forecasting**
>
> Although our paper focuses on how to improve the time series diffusion model, we also agree that it's necessary to include SOTA time series forecasting methods as a reference. Therefore, we included Autoformer[1], Non-stationary Transformer (NSformer)[2], and PatchTST[3] to compare deterministic forecasting performance. We run all models in the same setting in our paper, i.e., L  = {96, 192, 336, 720}.
>
> |            | Electricity | ETTh2 | ETTm2 | Exchange | traffic | weather |     rank |
> | :--------- | ----------: | ----: | ----: | -------: | ------: | ------: | -------: |
> | Autoformer |       0.594 | 0.218 | 0.168 |    0.601 |   0.267 |   0.293 |     3.83 |
> | NSformer   |       0.367 | 0.230 | 0.146 |    0.440 |   0.229 |   0.278 |     2.83 |
> | PatchTST   |       0.412 | 0.202 | 0.122 |    0.500 |   0.179 |   0.189 |     1.83 |
> | **ours**   |       0.340 | 0.212 | 0.124 |    0.372 |   0.166 |   0.207 | **1.50** |
>
> Regarding overall performance, our model still ranks first among these benchmarks, though it is slightly inferior to ETTh2 and weather compared to PatchTST.
>
> It should be noted that these SOTA architectures are particularly tailored for time series forecasting and well adapted to the benchmark datasets, while forecasting is one of the downstream applications of our proposed MA-TSD framework. Therefore, we think there exists great potential to accommodate the SOTA architectures into our MA-TSD framework to have a better forecasting performance in our future work.
>
>
> [1] Autoformer: Decomposition Transformers with Auto-Correlation for Long-Term Series Forecasting
>
> [2] Non-stationary Transformers: Exploring the Stationarity in Time Series Forecasting
>
> [3] A Time Series is Worth 64 Words: Long-term Forecasting with Transformers

---

### Official Review · Reviewer_mCdS · 2025-03-12

**Overall Recommendation:** 4

**Summary:**

When training a standard diffusion model on time series dataset, (Contribution 1) the authors identified that gradients conflict between small t and large t values, which hinders training. To address this issue, they propose (Contribution 2) a heuristic solution by adding "moving average" as an additional corruption to the forward equation, which is widely used in time series data. Additionally, since mean and variance play important roles in time series data, they trained the model for post-denormalization from z to x separately. They also propose an accelerated sampling method similar to DDIM.

Experiments were conducted on six time series datasets, (Contribution 3) showing the highest performance compared to baseline methods in forecasting and super-resolution tasks. Ablation studies demonstrate that the proposed methods significantly contributed to these performance improvements.

**Claims And Evidence:**

There are two claims in this paper:

(1) When training diffusion models on time series data, gradient conflicts occur between small timesteps and large timesteps.
(2) This issue can be mitigated through moving average data degradation.

And Claims 1 and 2 are supported phenomenologically through Figure 2.

However, the explanation of why MA-TSD prevents gradient conflict, i.e., Claim 2, still seems insufficient. Regarding the methodology of corruption through moving average, it would be much better if they could provide a more developed theoretical or intuitive motivation through toy examples beyond "time series data has more important low-frequency signals compared to other data."

(Minor Q) To my knowledge, gradient conflict phenomena and instability issues based on timestep regimes also occur in image datasets [1]. However, blurring diffusion models [2], which is similar method from this paper, considered do not effective as demonstrated in this paper. Could the reasons be: (1) moving average is more effective in time series data? (2) learning in latent space through VAE solves this problem? I'm curious about the inconsistency where similar methods weren't effective for images but achieved significant performance improvements in time series data. (3) Or is it because the noise schedule is determined by Eq. (13)? I'm curious about your opinion or intuition. Experimental evidence is plus.

[1] Truncated Consistency Models, https://arxiv.org/abs/2410.14895, ICLR25

[2] Blurring Diffusion Models, https://arxiv.org/abs/2209.05557

**Essential References Not Discussed:**

N/A

**Ethical Review Flag:**

Flag this paper for an ethics review.

**Experimental Designs Or Analyses:**

There are no special issues with the experimental settings in this paper.

A notable point is that Table 4's ablation study accurately shows the performance contribution size of moving average and instance normalization. I personally find the performance contribution of MA impressive.

**Methods And Evaluation Criteria:**

MA-TSD has methodological similarities to blurring diffusion models, but has (1) a distinct motivation regarding "gradient conflict" and (2) shows notable performance improvements in time series data, thus having distinguishable contributions.

Evaluation was conducted on six datasets using MSE and CRPS for time series forecasting, and Consistency and Context-FID for super-resolution evaluation. Performance improvements were shown across almost all datasets and metrics. Though I'm not an expert in this field, they conducted sufficiently extensive evaluations and demonstrated impressive performance improvements.

**Other Comments Or Suggestions:**

I hope to see qualitative results from various methods, M3VAE, TMDM, MR-DIFF, to understand the qualitative difference between various methods.

**Other Strengths And Weaknesses:**

I think the writing and the presentation is quite clear.

As I've mentioned several times, the greatest value of this paper is showing that a method I thought had been abandoned can still be valid in other domains.

**Questions For Authors:**

Q1: What exactly is the difference from Blurring Diffusion Models [1]? I'm curious about the points where the contribution is distinguished. If I can clearly understand where the contribution is distinct, I would like to raise my score to accept.

Q2: The fact that moving average aligns gradient signals by timestep seems like a very surprising discovery to me. Do you have any theory or intuition that can explain this? The reason that "time series data often have more informative low-frequency component" doesn't seem sufficient to me.

[1] Blurring Diffusion Models, https://arxiv.org/abs/2209.05557

**Relation To Broader Scientific Literature:**

Research that achieves performance improvements through changes to the forward equation has actually become an almost abandoned research direction in images, so it's impressive that it has been rediscovered in time series data. This shows that even if it wasn't effective for images, there is room for more research on forward equations for video or audio that have time series characteristics. Some might evaluate this research's novelty as low, but I'd like to assign high novelty in terms of rediscovery.

**Theoretical Claims:**

There are no theoretical claims in this paper that require verification.

---

> ### Author Rebuttal · Authors · 2025-03-31
>
> We greatly appreciate your acknowledgement of our work. We would like to address your questions as follows.
>
> **Q1: The difference between Blurring Diffusion Model (BDM) and ours**
>
> From a high-level perspective, BDM and ours shared a similar idea, i.e. building the degradation process with low-pass filters (blurring in BDM, MA in ours). However, there exists clear distinctions.
>
> 1. **Filtering space**: Though we think it's minor due to convolution theoram, we still point out at first to pave the way for the following parts. Specifically, we filtered the data in the time space by convolution (matrix multiplication),$q(\mathbf{x}_t | \mathbf{x}_0) = \mathcal{N} (\mathbf{K}_t\mathbf{x}_0,\beta_t^2 \mathbf{I})$ while BDM blurs the images in the frequency domain and transform back to the pixel domain, i.e. $q(\mathbf{x}_t|\mathbf{x}_0) = \mathcal{N} ( \mathbf{V} \boldsymbol{\alpha}_t \mathbf{V}^\top \mathbf{x}_0, \sigma_t^2 \mathbf{I})$, where $\mathbf{V}^\top,\mathbf{V}$ are  DCT and IDCT, and $\boldsymbol{\alpha}_t$ is the frequency response of Gaussian blurring kernel, a diagnoal matrix, whose each entry $\alpha_t^i \in (0, 1]$ is a coefficient of i th frequency component. For low pass filters, $\alpha_t^i$ decreases by i until (nearly) zero to suppress high frequency components.
>
> 2. **Markovian or not (Major)**: Since $\boldsymbol{\alpha_t}$ is diagnoal, BDM proposed that for each frequency component, a standard Markovian DDPM can be constructed. The one-step transition is accordingly defined, i.e. $q(u_t | u_s) = \mathcal{N}(\alpha_{t|s}u_s, \sigma_{t | s}^2)$, where s=t-1, $u_t=V^\top x_0$ is the frequency representation and $\boldsymbol{\alpha}_{t | s}=\boldsymbol{\alpha}_t / \boldsymbol{\alpha}_s$. However, dividing  $\boldsymbol{\alpha}_s$ could be problematic in practice, because $\alpha_t^i$ could become to be (nearly) zero for large i, so dividing $\boldsymbol{\alpha}_s$ is numerically unstable for all diffusion steps. Therefore, though BDM claimed to have Markov transition under the DDPM framework, we think it's improper to define $q(x_t | x_s)$ when the transition operation from $x_0$ to $x_t$ is non-invertible. In our framework, faced with similar invertible MA, we bypassed the definition of $q(x_t|x_s)$, assumed $q(x_1:T | x_0)$ non-Markovian (in the DDIM-style) and then delicately defined $q(x_s|x_t, x_0)$ to satisfy $q(x_t|x_0)$ for all t. Thus, whether it's Markovian or not is another distinct difference from ours and BDM.
>
> 3. **Noise schedule**: BDM designed $\boldsymbol{\alpha}_t = a_t \mathbf{d}_t$, where $\mathbf{d}_t$ is the frequency response of blurring kernel and $a_t \in [0,1]$ is an extra scaler decreasing by t, and the noise schedule $\sigma_t = 1-a_t^2$. In our framework, the noise schedule $\beta_t$ is dataset-based, chosen regarding the variance decrease caused by MA (Eq.13) on different datasets. To compare these two, we mimic BDM's noise schedule, i.e. multiplying an $a_t$ to $\mathbf{K}_t$ and set $\beta_t = 1-a_t^2$, and ran the experiments on forecasting task (L=96). We can see that test errors indeed increased, which might be because of BDM's $\sigma_t$ schedule ignored the extra effects of blurring ($\mathbf{d}_t$) and only consider the $a_t$.
>
> |   Dataset   |  Model   |  MSE  | CRPS  |
> | :---------: | :------: | :---: | :---: |
> | electricity |  ours  | 0.288 | 0.158 |
> | | ours+a | 0.303 | 0.158 |
> |    ETTh2    |  ours  | 0.136 | 0.121 |
> | | ours+a | 0.166 | 0.130 |
> |  Exchange   |  ours  | 0.098 | 0.110 |
> | | ours+a | 0.193 | 0.136 |
>
> In summary, despite the similar high-level idea, there exist clear differences between BDM and ours. BDM tried to fit in the standard DDPM framework, while we reformulated a framework with special adaptation on moving average filters and time series. Unfortunately, BDM didn't release their official codes, and we are unable to reproduce it on time series data for more quantitative comparison.
>
> **Q2: Moving average aligns gradient signals**
>
> By intuition, a mild data degradation in diffusion should benefit the learning of denoising networks, since the inputs could be mostly noise if degradation is too fast. For example T=500 in Fig.2, the gradients at t> ~190 (right down block) are similar because x_t are almost noise and the model can hardly learn anything. For t < ~190 (left up block), they are more informative and contribute to denoising learning more. Thus, we'd like to "prolong" the left up block to help the learning.
>
> Meanwhile, to reconstruct a time series, low-frequency (LF) information is usually more helpful. Thus, we're motivated to use MA, a widely used TS low-pass filter, to design a mild degradation where LF is kept more and serves as common information for all diffusion steps. In this way, the inputs for denoising models are more structurally similar so that gradients conflicts at different steps could be less.
>
> **Others**: Forecasting examples are in https://anonymous.4open.science/r/ICML2025_rebuttal-134C/forecast_plot.png

---

> > ### Comment · Reviewer_mCdS · 2025-04-04
> >
> > I appreciate the author's hard work. I had questions about the exact differences between BDM and your formulation, and what distinct contributions each has, but those parts have been clarified. Accordingly, I'm raising my score to **accept**.
> >
> > > Q1: The difference between Blurring Diffusion Model (BDM) and yours
> >
> > Thank you for the explanation. To summarize my understanding:
> >
> > - Thanks to non-Markovian properties: (1) You can avoid explicitly defining q(x_t|x_0) which has numerical stability issues, and (2) The formulation can work even when the corruption from x_0 to x_t is not invertible. (But, actually, I still don't fully understand exactly what the advantage of point 2 is)
> >
> > - Thanks to data-dependent noise schedule: Performance improvement
> >
> > These are the two differences from BPM. I now understand which parts are similar and which are different, and accordingly, I'll adjust my score upward. It would be good to briefly mention this somewhere in the paper. Rather than emphasizing that there's a distinct contribution, it would be better to explain what changes were necessary when applying BDM's ideas to time series.
> >
> > > Q2: Moving average aligns gradient signals
> >
> > There's a logical jump, but it's an interesting perspective.
> >
> > First, regarding the amount of information learned at each timestep, [1] proposes a noise schedule that makes the amount of information to be learned at each timestep even. This research seems to suggest that simply adjusting the noise schedule is insufficient and that it also depends on the characteristics of the information contained in the data. I find your intuition quite convincing, and it's an interesting intuition that explains why your method works well.
> >
> > However, just because the information learned at each timestep is evenly distributed doesn't necessarily mean that gradient conflicts won't occur. This is because the types of information learned at each timestep can differ. In fact, in Figure 2 left down, it appears that the gradient from small diffusion timesteps is interfering with other regions. This seems somewhat inconsistent with the intuition given that it's an area with sufficient remaining information.
> >
> > Nevertheless, the intuition that if information is removed too quickly, proper learning doesn't occur in later timesteps resulting in meaningless gradients, seems reasonable. Thank you for the explanation.
> >
> > [1] Continuous diffusion for categorical data, https://arxiv.org/abs/2211.15089

---

> > > ### Author Response · Authors · 2025-04-05
> > >
> > > We greatly appreciate your insightful and valuable discussion on both questions, as well as your acknowledgment of our paper. For your further comments, we'd like to explain a bit more about the effects of non-Markovian design and have some general discussion on the gradient conflicts.
> > >
> > > **1. Non-Markovian design**
> > >
> > > In fact, whether it's Markovian or not affects the formulation of backward process, i.e. $q(\mathbf{x}_{t-1} | \mathbf{x}_t, \mathbf{x}_0)$.
> > >
> > > In the BDM case, the DDPM-like Markovian assumption can simplify the calculation of $q(\mathbf{x}_{t-1} | \mathbf{x}_t, \mathbf{x}_0)$, but requires the formulation of  q(x_t | x_0) and  q(x_t | x_t-1). If we'd like to define the q(x_t | x_0) with non-invertible low pass filters, then the q(x_t | x_t-1) will be badly defined because it involves inverting blurring kernels (Eq.16-18, 21 in BDM), which is practically unstable. Though some epsilons can be added to ensure the codes "runable", tiny errors in the high frequency components will still be amplified a lot through the iterative backward process. Chances are that the generated data are dominated by the improperly amplified high frequency components.
> > >
> > > In our case, we bypassed the Markovian assumption, so the formulation of q(x_t | x_t-1) is also skipped. Instead, we formulate $q(\mathbf{x}_{t-1} | \mathbf{x}_t, \mathbf{x}_0)$ in a DDIM style as long as it satisfies q(x_t | x_0). The resulting backward equation (Eq. 14, in our paper) doesn't involve any inverting operation of MA kernels. Thus, we can essentially avoid such problem with the non-Markovian design.
> > >
> > > We also highly agree that it would be better to mention these differences in the paper, and we will thoroughly integrate your suggestions into our revised manuscript.
> > >
> > > **2. Gradient conflicts**
> > >
> > > Thanks for your insightful opinion on this problem. According to our understanding of your opinion, gradient conflicts may come from two parts (1) the uneven information distribution at different steps and (2) the differences in the information that the denoising network tries to learn.
> > >
> > > Indeed, our intuition mainly came from the focus on the first part. Accordingly, our framework is also aimed for alleviating the gradient conflicts caused by the uneven information distribution. We totally agree that analyzing the joint contribution to gradient conflicts of both parts could help us have a deeper understanding on diffusion training, and we will keep investigating on such topic in our future works.

---

### Official Review · Reviewer_gWYs · 2025-03-12

**Overall Recommendation:** 3

**Summary:**

This paper presents a non-isotropic time series diffusion model (MA-TSD) for time series analysis. The key idea is to use a moving average in the forward process to better preserve low-frequency information, thereby avoiding gradient conflicts during training. The model also features an accelerable backward process, similar to DDIM, which can be viewed as time series super-resolution, and employs instance normalization to standardize the data. Experiments on real-world datasets (such as Electricity, ETTh2, and Traffic) show that MA-TSD outperforms existing diffusion-based models in both forecasting and super-resolution tasks.

**Claims And Evidence:**

The paper argues that using standard DDPM on time series causes gradient conflicts because low-frequency information decays too quickly, leading to inconsistent data perception across diffusion steps. It claims that incorporating a moving average in the forward process helps preserve low-frequency components, resulting in more stable training and higher-quality generation. These claims are backed by both theoretical analyses (such as gradient similarity matrices and spectral energy ratio comparisons) and experimental evidence on datasets like Electricity. However, the comparisons are mainly limited to diffusion models and do not include mainstream non-diffusion methods like Transformers or RNNs.

**Essential References Not Discussed:**

No.

**Experimental Designs Or Analyses:**

The experiments are well-designed. The authors tested their model on six real-world datasets with different time patterns, using metrics such as MSE, CRPS, and Context-FID, and they performed ablation studies to evaluate the key components of MA-TSD. The results clearly demonstrate the model’s advantages in training stability, low-frequency information retention, and generation efficiency. Including comparisons with non-diffusion models could further improve the evaluation.
Please note that my understanding of time series' research is limited, so some of my comments might not be completely accurate.

**Methods And Evaluation Criteria:**

The method combines a moving average forward process, instance normalization, and an accelerable backward process to better handle time series frequencies and preserve low-frequency information. It is evaluated on forecasting and super-resolution tasks using metrics like MSE, CRPS, and Context-FID. Overall, these techniques and metrics are suitable for the problem, though the model is complex and might be computationally demanding for large datasets. (Note: My background in time series is limited, so I may not fully grasp all the details of these evaluation metrics.)

**Other Comments Or Suggestions:**

Nothing for this part.

**Other Strengths And Weaknesses:**

Strengths:
- The model demonstrates superior performance in both time series forecasting and super-resolution tasks.
- The work combines rigorous theoretical analysis with comprehensive experimental validation, lending strong support to its claims.

Weaknesses:
- Despite the accelerated backward process, the overall model structure is complex, and the computational demands may be high when processing large-scale time series data.
- The experiments are primarily conducted on specific datasets, so the model’s performance on other types or domains of time series data remains to be further verified.

Again, I am not an expert in time series.

**Questions For Authors:**

No, just reply the weakness.

**Relation To Broader Scientific Literature:**

The paper builds on established diffusion models like DDPM and DDIM, adapting them for time series by using a moving average to preserve low-frequency information and improve training stability. This approach extends ideas from image and video synthesis to sequential data tasks such as forecasting and super-resolution. Please note that I am not a time series expert, so my understanding of the broader literature in this area is limited.

**Theoretical Claims:**

The paper presents clear and rigorous derivations for the forward and backward processes, noise scheduling, and the conditional joint distribution. These proofs are based on fundamental principles of probability and mathematics. Although the derivations can be dense for non-experts, no major errors were found.

---

> ### Author Rebuttal · Authors · 2025-03-31
>
> We sincerely appreciate your time and effort in reviewing, as well as your constructive feedback, which helps strengthen our work.  We would like to address your questions as follows.
>
> **1. Lack of non-diffusion benchmarks for comparison**
>
> We agree that non-diffusion benchmarks are also necessary for general comparison, even if our work focuses on the improvement of time series diffusion models. We've added related experiments with Autoformer, Non-stationary transformer, and PatchTST for time series forecasting.
>
>
> |            | Electricity | ETTh2 | ETTm2 | Exchange | traffic | weather |     rank |
> | :--------- | ----------: | ----: | ----: | -------: | ------: | ------: | -------: |
> | Autoformer |       0.594 | 0.218 | 0.168 |    0.601 |   0.267 |   0.293 |     3.83 |
> | NSformer   |       0.367 | 0.230 | 0.146 |    0.440 |   0.229 |   0.278 |     2.83 |
> | PatchTST   |       0.412 | 0.202 | 0.122 |    0.500 |   0.179 |   0.189 |     1.83 |
> | **ours**   |       0.340 | 0.212 | 0.124 |    0.372 |   0.166 |   0.207 | **1.50** |
>
> Regarding overall performance, our model still ranks first among these benchmarks, though it is slightly inferior to ETTh2 and weather compared to PatchTST.
>
> It should be noted that these SOTA architectures are particularly tailored for time series forecasting and well adapted to the benchmark datasets, while forecasting is one of the downstream applications of our proposed MA-TSD framework. Therefore, we think there exists great potential to accommodate the SOTA architectures into our MA-TSD framework to have a better forecasting performance in our future work.
>
> **2. Computational demands**
>
> We'd like to analyze the computation burden of our method from two perspectives, i.e., spatially and temporally.
>
> Spatially: It should be noted again that the training process of our framework is almost identical to the standard diffusion, i.e., sample a batched time series window, degrade it, and use a denoising network to learn. The main difference lies in the forward (degrading) process. In our framework, we need to store a series of kernel matrices $\boldsymbol{K}_t$ in advance for degrading at different diffusion time steps, which in practice is a tensor with shape $T \times L \times L$ (T=diffusion steps, L=sequence length). Thus, if we try to model an extremely long sequence, it could increase the storage demand in quadratic growth. Fortunately, $\boldsymbol{K}_t$ is sparse because only unconvolved time steps remain zeros, and such sparsity decreases by t. For example, when $T=100, L=576$, the sparsity ($1 - oc / L^2$, $oc$ is the number of non-zero entries) at $t=20,40,60,80$ is [0.979 0.945 0.879 0.694]. Therefore, though we need to store a series of kernel matrices, they are mostly highly sparse, thus not imposing much storage burden.
>
> Temporally: The most time-consuming part of diffusion models is sampling, but we can utilize the accelerated backward process to speed up, especially in the time series super-resolution task. As we illustrated in the paper, we conduct SR naturally through our backward process without retraining the whole model like benchmarks that rely on conditional inputs.
>
> In practice, we used only one NVIDIA 4090 24GB to conduct all the experiments. Even if we run a $L=720$ task from scratch on the largest dataset (ettm2), it only takes $\textbf{0.72}$ GPU hours to train and sample, thus no significant computational demands in our method.
>
> **3. Dataset diversity**
>
> We highly agree that dataset diversity is important for evaluation. For now, our used datasets have included Traffic (traffic system), Electricity/MFRED (power system), Exchange_rate (financial market), ETT (IoT sensors), Weather (meteorological system), and Solar/Wind (renewable energy). These are widely used benchmark datasets in the field of time series [1,2].
>
> Beyond these, we also added an ECG dataset (medical system) during the rebuttal period for a more complete evaluation. We launched a standard time series generation experiment on the ECG dataset and included two more benchmarks, i.e., KoVAE (RNN-based) and ImagenTime (Diffusion-based). The results are shown below.
>
> | Dataset  |   Model    | Disc. Score | Pred. Score | Context-FID |
> | :------: | :--------: | :---------: | :---------: | :---------: |
> | ECG |   KoVAE    |    0.459    |    0.081    |    1.206    |
> | | ImagenTime |    0.400    |    0.079    |    1.223    |
> | |  **ours**  |  **0.345**  |  **0.076**  |  **0.979**  |
>
> Disc. Score evaluates the fidelity by training a post-classifier to tell whether the data is real or not. Pred. Score evaluates the usefulness by training an RNN forecasting model with the generated data and testing it on the real [3]. Thus, we can see the superiority of our method on both fidelity and usefulness over benchmarks.
>
> [1] Deep Time Series Models: A Comprehensive Survey and Benchmark
>
> [2] A Survey on Diffusion Models for Time Series and Spatio-Temporal Data
>
> [3] TSGBench: Time Series Generation Benchmark

---

### Decision · Program_Chairs · 2025-05-01

**Decision:**

Accept (poster)

**Comment:**

This paper receives two weak accept and two accept. It proposes to use moving average to maintain more informative low-frequency components which are important for time series. Even though all the reviewers consider acceptance, I encourage the authors to update the final version according to the review suggestions.